# On Uncertainty in Deep State Space Models for Model-Based Reinforcement Learning

**Philipp Becker**  *philipp.becker@kit.edu*
*Autonomous Learning Robots Lab, Karlsruhe Institute of Technology*

**Gerhard Neumann**  *gerhard.neumann@kit.edu*
*Autonomous Learning Robots Lab, Karlsruhe Institute of Technology*

**Reviewed on OpenReview:** *https://openreview.net/forum?id=UQXdQyoRZh*

## Abstract

Improved state space models, such as *Recurrent State Space Models (RSSMs)*, are a key factor behind recent advances in model-based reinforcement learning (RL). Yet, despite their empirical success, many of the underlying design choices are not well understood. We show that *RSSMs* use a suboptimal inference scheme and that models trained using this inference overestimate the aleatoric uncertainty of the ground truth system. We find this overestimation implicitly regularizes *RSSMs* and allows them to succeed in model-based RL. We postulate that this implicit regularization fulfills the same functionality as explicitly modeling epistemic uncertainty, which is crucial for many other model-based RL approaches. Yet, overestimating aleatoric uncertainty can also impair performance in cases where accurately estimating it matters, e.g., when we have to deal with occlusions, missing observations, or fusing sensor modalities at different frequencies. Moreover, the implicit regularization is a side-effect of the inference scheme and not the result of a rigorous, principled formulation, which renders analyzing or improving *RSSMs* difficult. Thus, we propose an alternative approach building on well-understood components for modeling aleatoric and epistemic uncertainty, dubbed *Variational Recurrent Kalman Network (VRKN)*. This approach uses Kalman updates for exact smoothing inference in a latent space and Monte Carlo Dropout to model epistemic uncertainty. Due to the Kalman updates, the *VRKN* can naturally handle missing observations or sensor fusion problems with varying numbers of observations per time step. Our experiments show that using the *VRKN* instead of the *RSSM* improves performance in tasks where appropriately capturing aleatoric uncertainty is crucial while matching it in the deterministic standard benchmarks[1].

## 1 Introduction

Accurate system models are crucial for model-based control and reinforcement learning (RL) in autonomous systems applications under partial observability. Practitioners commonly use state space models (SSMs) (Murphy, 2012) to formalize such systems. SSMs consist of a dynamics model, describing how one state relates to the next, and an observation model, which describes how system states generate observations. Yet, dynamics and observation models are unknown for most relevant problems, and exact inference in the resulting SSM is usually intractable. Researchers have proposed numerous approaches to learn the models from data and approximate the inference to solve those issues.

*Recurrent State Space Models (RSSMs)* (Hafner et al., 2019) are of particular interest here. Using *RSSMs* as the backbone for their *Deep Planning Network (PlaNet)*, Hafner et al. (2019) showed that variational latent dynamics learning can succeed in image-based RL for complex control tasks. Combined with simple

---

[1]Code available at: `https://github.com/pbecker93/vrkn`

planning, *RSSMs* can match the performance of model-free RL while requiring significantly fewer environment interactions. The authors later improved upon their original model, including a parametric policy trained on imagined trajectories (*Dreamer*) (Hafner et al., 2020). In general, approaches based on *RSSM*s have found considerable interest in the model-based RL community. Yet, while *RSSM*s draw inspiration from classical SSMs, they use a simplified inference scheme. During inference, they assume the belief is independent of future observations instead of using the correct smoothing assumptions (Murphy, 2012) to obtain the belief. We formalize this observation in Section 2 and discuss how these simplified assumptions result in a theoretically looser variational lower bound. Further, we analyze the effects of these assumptions on model learning and find they cause an overestimation of aleatoric uncertainty. Such aleatoric uncertainty stems from partial observability or inherent stochasticity of the system (Hüllermeier & Waegeman, 2021). It plays an important role in many realistic tasks, e.g., in the form of occlusions, missing observations, or observations arriving in different modalities and frequencies. Consider, for example, low-frequency camera images providing external information and high-frequency proprioceptive measurements of the robot's internal state. Given such sensory inputs, we require an appropriate estimation of aleatoric uncertainty to fuse all information optimally and form accurate belief states. Our experiments show that *RSSMs* performance sub-optimally in such cases, which we attribute to the wrong estimation of aleatoric uncertainty.

Furthermore, we argue that the *RSSM's* inference assumptions are a double-edged sword for model-based RL. On the one hand, the overestimated aleatoric uncertainty can be beneficial as it leads to dynamics models that generalize better and are more robust to *objective mismatch* (Luo et al., 2019; Lambert et al., 2020). Such *objective mismatch* arises because the model aims to maximize the data likelihood while the metric we care about is the agent's reward. Additionally, as the agent explores during training, it will encounter states and actions the model has not seen, causing a distribution shift between training and data collection. While many approaches rely on explicit epistemic uncertainty to tackle this issue (Chua et al., 2018; Janner et al., 2019), *RSSMs* succeed without capturing epistemic uncertainty. Such epistemic uncertainty (Hüllermeier & Waegeman, 2021) stems from the lack of training data and plays an important role in model-based RL, where the data is not given but collected during the training process. On the other hand, this heuristic approach to address *objective mismatch* complicates the design and analysis of the approach as the robustness is a side-product of the inference scheme and does not follow well-motivated principles. For example, purely stochastic versions of the *RSSM* gives unsatisfactory results, and it is unclear why this is the case. Arguably, an explanation could be the overestimated aleatoric uncertainty causing the purely stochastic model to be unstable. As a remedy, Hafner et al. (2019) introduce a *deterministic path*, i.e., they combine stochastic and deterministic features to form the belief state.

We show that removing this issue for *RSSMs* by implementing a naive approach to smoothing yields unsatisfactory results, even if the model uses explicit epistemic uncertainty to compensate for the missing overestimation of the aleatoric uncertainty. To make smoothing inference work, we redesign the model using well-understood and theoretically founded components to model aleatoric and epistemic uncertainty. We dub the resulting model *Variational Recurrent Kalman Network (VRKN)*. It uses a linear Gaussian State Space Model embedded in a latent space, which allows closed-form smoothing inference and proper estimation of aleatoric uncertainty. Furthermore, as the inference builds on Bayes rule, it provides a natural treatment of missing observations and sensor fusion settings. Additionally, the *VRKN* uses Monte Carlo Dropout (Gal & Ghahramani, 2016) for a Bayesian treatment of its transition model's parameters, explicitly modeling epistemic uncertainty.

We first show that the resulting architecture allows model-based agents to perform comparably to *RSSM*-based agents on standard benchmarks from the Deep Mind Control Suite (Tassa et al., 2018). Those environments are almost deterministic, i.e., the aleatoric uncertainty is low. Subsequently, we modify tasks from the Deep Mind Control Suite to create three different scenarios which exhibit high aleatoric uncertainty, i.e., (i) we introduce occlusions, (ii) missing observations, and (iii) sensor fusions problems of external camera images and internal proprioceptive feedback which are available at different frequencies. Using those, we show that the *VRKN* improves the agents' performance due to its accurate estimation of aleatoric uncertainty. Our approach builds on well-motivated components for aleatoric and epistemic uncertainty. Additionally, it does not require heuristic model components such as a *deterministic path*. As such, it may serve as a basis for future research into improving state space models for model-based RL.

To summarize our contributions:

1. We analyze the assumptions underlying the *RSSM* and show that they are not only suboptimal but also have subtle effects on model learning. They lead to an overestimation of the aleatoric uncertainty (Section 2.1 and Section 2.2).

2. We argue that, counterintuitively, this suboptimal inference is beneficial for the *RSSMs* performance as it addresses *objective mismatch* in a heuristic way (Section 2.3). We evaluate several modifications to the *RSSM* to provide empirical evidence to support this hypothesis (Section 4.1).

3. We introduce the *VRKN*, providing a more principled inductive bias for smoothing inference than the *RSSM* (Section 3). We again show that a smoothing inference without additional measures results in suboptimal performance, yet, when combined with epistemic uncertainty, the *VRKN*'s improved inductive bias allows it to close the performance gap on standard benchmarks (Section 4.1).

4. We show that *VRKN*-based agents improve performance in tasks where correct uncertainty estimation matters. Here we consider tasks with partial observability, missing information, or tasks that require sensor fusion. (Section 4.2)

## 2   Inference and Learning in State Space Models

State Space Models (SSMs) (Murphy, 2012) assume that a sequence of observations $\mathbf{o}_{\leq T} = \{\mathbf{o}_t\}_{t=0\cdots T}$ is generated by a sequence of latent state variables $\mathbf{z}_{\leq T} = \{\mathbf{z}_t\}_{t=0\cdots T}$, given a sequence of actions $\mathbf{a}_{\leq T} = \{\mathbf{a}_t\}_{t=0\cdots T}$. In SSMs, each observation $\mathbf{o}_t$ is assumed to only depend on the current latent state $\mathbf{z}_t$ via an observation model $p(\mathbf{o}_t|\mathbf{z_t})$. Further, they assume the latent states are Markovian, i.e., each latent state only depends on its direct predecessor and the corresponding action via a dynamics model $p(\mathbf{z}_{t+1}|\mathbf{z}_t, \mathbf{a}_t)$. Finally, the initial state is distributed according to a distribution $p(\mathbf{z}_0)$. Figure 1a shows the corresponding graphical model. Typically, when inferring latent states from observations, we consider three different beliefs for each $\mathbf{z}_t$. Those are the prior $p(\mathbf{z}_t|\mathbf{o}_{\leq t-1}, \mathbf{a}_{\leq t-1})$, i.e., the belief before observing $\mathbf{o}_t$, the posterior, $p(\mathbf{z}_t|\mathbf{o}_{\leq t}, \mathbf{a}_{\leq t-1})$, i.e., the belief after observing $\mathbf{o}_t$, as well as the smoothed belief $p(\mathbf{z}_t|\mathbf{o}_{\leq T}, \mathbf{a}_{\leq T})$ which is conditioned on all future observations and actions, until the last time step $T$. We refer to those estimates as state-beliefs to distinguish them from dynamics distributions, which are con-

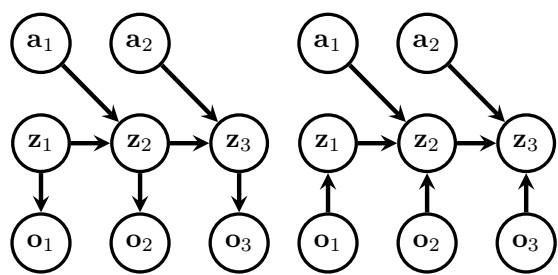

(a) State Space Model    (b) *RSSM* Inference Model

Figure 1: (a) State space model, serving as the generative model throughout this work. (b) Graphical Model underlying the *RSSM* (Hafner et al., 2019) inference scheme. In contrast to the generative SSM, the direction between observations and latent states is inverted. These independence assumptions result in a simplified inference and subtle effects on model learning.

ditioned on the previous state $\mathbf{z}_{t-1}$ such as the prior dynamics $p(\mathbf{z}_t|\mathbf{z}_{t-1}, \mathbf{a}_{t-1})$, the posterior dynamics $p(\mathbf{z}_t|\mathbf{z}_{t-1}, \mathbf{o}_t, \mathbf{a}_{t-1})$ and the smoothed dynamics $p(\mathbf{z}_t|\mathbf{z}_{t-1}, \mathbf{o}_{\geq t}, \mathbf{a}_{\geq t-1})$. To get from a dynamics distribution to the corresponding state belief we need to marginalize out the previous state, e.g., $p(\mathbf{z}_t|\mathbf{o}_{\leq t-1}, \mathbf{a}_{\leq t-1}) = \int p(\mathbf{z}_t|\mathbf{z}_{t-1}, \mathbf{a}_{t-1})p(\mathbf{z}_{t-1}|\mathbf{o}_{\leq t-1}, \mathbf{a}_{\leq t-1})d\mathbf{z}_{t-1}$ for the prior.

Traditionally, the independence assumptions of the generative model, shown in Figure 1a, are also used for inference. Yet, *RSSMs* (Hafner et al., 2019) work with a different set of assumptions during inference, shown in Figure 1b[2]. In particular, in this graphical model the state $\mathbf{z}_t$ is conditionally independent of all observations $\mathbf{o}_{>t}$ and actions $\mathbf{a}_{\geq t}$ given $\mathbf{z}_{t-1}$, $\mathbf{a}_{t-1}$, and $\mathbf{o}_t$, which is not the case for the standard SSM. We

---

[2]The full model of Hafner et al. (2019) also includes a *deterministic-path* which is of no concern regarding the discussion here. Thus, we omit it for brevity. Furthermore, Hafner et al. (2019) compare their *RSSM* to a baseline they abbreviate as *SSM* (stochastic state model), which also builds on the simplified assumptions. In this work, we refer to all approaches based on the simplified assumptions as *RSSM*.

discuss the effects of those assumptions on inference and model learning. We refer to Appendix A for more detailed derivations of the following identities.

## 2.1 Variational Inference for State Space Models

Inferring beliefs over the latent states given observations and actions is intractable for most models. Thus, we usually use approximate inference methods. In this work, we focus on variational inference. Such variational methods introduce an auxiliary distribution $q$ over the latent variables given the observable variables. They decompose the data log-likelihood into a variational lower bound and a Kullback-Leibler divergence (KL) term, measuring how close the bound is to the data log-likelihood. For State Space Models (SSMs) this decomposition is given as $\log p(\mathbf{o}_{\leq T}|\mathbf{a}_{\leq T}) =$

$$\underbrace{\mathbb{E}_{q(\mathbf{z}_{\leq T}|\mathbf{o}_{\leq T},\mathbf{a}_{\leq T}))}\left[\log \frac{p(\mathbf{o}_{\leq T}, \mathbf{z}_{\leq T}|\mathbf{a}_{\leq T})}{q(\mathbf{z}_{\leq T}|\mathbf{o}_{\leq T},\mathbf{a}_{\leq T})}\right]}_{\text{Lower Bound}} + \underbrace{\mathrm{KL}\left[q(\mathbf{z}_{\leq T}|\mathbf{o}_{\leq T},\mathbf{a}_{\leq T}) \parallel p(\mathbf{z}_{\leq T}|\mathbf{o}_{\leq T},\mathbf{a}_{\leq T})\right]}_{\text{KL term}(\geq 0)}. \tag{1}$$

Variational inference methods try to find the optimal $q$ by maximizing the lower bound or minimizing the KL term. If a $q$ can be found such that $q(\mathbf{z}_{\leq T}|\mathbf{o}_{\leq T},\mathbf{a}_{\leq T}) = p(\mathbf{z}_{\leq T}|\mathbf{o}_{\leq T},\mathbf{a}_{\leq T})$, the KL term is 0, and the bound is said to be tight. While the decomposition is valid for arbitrary distributions $q(\mathbf{z}_{\leq T}|\mathbf{o}_{\leq T},\mathbf{a}_{\leq T})$, we need to pose a set of independence assumptions to obtain tractable variational models. If we again use the independence assumptions of the generative model, shown in Figure 1a, we can obtain the inference model by explicitly inverting the generative direction using Bayes rule, $q(\mathbf{z}_{\leq T}|\mathbf{o}_{\leq T},\mathbf{a}_{\leq T}) = \frac{1}{p(\mathbf{o}_{\leq T})}p(\mathbf{z}_0)\prod_{t=1}^{T} p(\mathbf{z}_t|\mathbf{z}_{t-1},\mathbf{a}_t)\prod_{t=0}^{T} p(\mathbf{o}_t|\mathbf{z}_t)$. For certain model parametrizations, this variational distribution can be computed analytically, e.g., by Kalman smoothing if the model is linear and Gaussian. In this case, the KL term vanishes, and the bound is tight.

Yet, *RSSMs*, as introduced in (Hafner et al., 2019), assume the variational distribution factorizes as

$$q(\mathbf{z}_{\leq T}|\mathbf{o}_{\leq T},\mathbf{a}_{\leq T}) = q(\mathbf{z}_0|\mathbf{o}_0)\prod_{t=1}^{T} q(\mathbf{z}_t|\mathbf{z}_{t-1},\mathbf{o}_t,\mathbf{a}_{t-1}).$$

This assumption results in a simplified inference procedure, as the belief over $\mathbf{z}_t$ is assumed to be independent of all future observations $\mathbf{o}_{>t}$, given $\mathbf{z}_{t-1}$, $\mathbf{o}_t$, and $\mathbf{a}_{t-1}$.

Inserting this assumptions into the KL term yields $\mathrm{KL}\left[q(\mathbf{z}_{\leq T}|\mathbf{o}_{\leq T},\mathbf{a}_{\leq T}) \parallel p(\mathbf{z}_{\leq T}|\mathbf{o}_{\leq T},\mathbf{a}_{\leq T})\right] =$

$$\sum_{t=1}^{T} \mathbb{E}_{q(\mathbf{z}_{t-1}|\mathbf{o}_{\leq t-1},\mathbf{a}_{\leq t-2})}\left[\mathrm{KL}\left[q(\mathbf{z}_t|\mathbf{z}_{t-1},\mathbf{o}_t,\mathbf{a}_{t-1}) \parallel p(\mathbf{z}_t|\mathbf{z}_{t-1},\mathbf{o}_{\geq t},\mathbf{a}_{\geq t-1})\right]\right]. \tag{2}$$

For general distributions $p$, the individual terms in this sum will not be 0, as the right-hand side distributions receive information, i.e., future observations and actions, which are ignored by the variational distribution. Thus, this bound can in general not be tight, not even for linear Gaussian models as the resulting variational distribution can in general not represent $p(\mathbf{z}_{\leq t}|\mathbf{o}_{\leq T},\mathbf{a}_{\leq T})$. Typically, tight variational bounds are preferable as they allow for faster optimization of the marginal log-likelihood.

The above discussion may be hypothetical, as all considered architectures do not provide tight lower bounds due to the use of deep neural networks, which prevents analytic solutions for inference. Still, as a tight lower bound does not even theoretically exist for the *RSSM* assumptions, we believe this is already an indication of the misspecification of its inference distribution.

## 2.2 Model Learning under Different Inference Assumptions

In the model-based RL setting considered in this work, the generative model is usually assumed to be unknown and we jointly learn a parametric generative model $p_{\boldsymbol{\theta}}$ and an inference model $q_{\psi}$ using an auto-encoding variational Bayes approach (Kingma & Welling, 2013; Sohn et al., 2015) by maximizing the lower

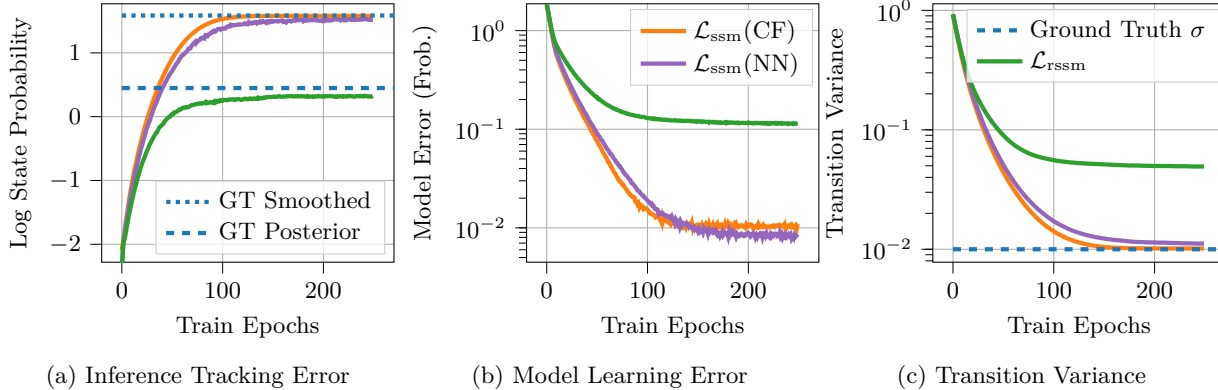

(a) Inference Tracking Error    (b) Model Learning Error    (c) Transition Variance

Figure 2: Comparison of inference and model learning results on a simple linear-Gaussian state space model without actions, under both *RSSM* and SSM inference assumptions. We report average results over 10 seeds and error bars indicating 95% bootstrapped confidence intervals. Note that the error bars are too small to be visible. We consider both a closed-form (CF) version based on Kalman Smoothing and a version with a neural network (NN) as an inference model for the SSM-based approaches and find that only the objective matters, not the parametrization of the inference model. **(a)** Log-probability of the ground truth states under the learned models. We compare against the quality of the ground truth smoothed (GT Smoothed) and posterior (GT Posterior) beliefs, computed using a Kalman Smoother and ground truth generative model. While the SSM objective reaches the quality of the smoothed belief, the *RSSM*-based inference fails to attain the quality of even the posterior belief. **(b)**: Distance between ground truth transition matrix and learned transition matrix, measured using the Frobenius norm. Here, the SSM inference yields a model that is an order magnitude closer to the ground-truth model than that learned by the *RSSM*-bound. **(c)**: Transition variance $\tilde{\sigma}\mathbf{I}$. With the SSM bound we recover the ground-truth aleatoric uncertainty, yet with the *RSSM* bound the aleatoric uncertainty is significantly overestimated.

bound part of Equation 1. Inserting the *RSSM*-assumptions into the general lower bound gives the objective introduced by Hafner et al. (2019), $\mathcal{L}_{\mathrm{rssm}}(\mathbf{o}_{\leq T}, \mathbf{a}_{\leq T}) =$

$$\sum_{t=1}^{T} \mathbb{E}_{q_{\boldsymbol{\psi}}(\mathbf{z}_t|\mathbf{o}_{\leq t},\mathbf{a}_{\leq t})}\left[\log p_{\boldsymbol{\theta}}(\mathbf{o}_t|\mathbf{z}_t)\right] - \mathbb{E}_{q_{\boldsymbol{\psi}}(\mathbf{z}_{t-1}|\mathbf{o}_{\leq t-1},\mathbf{a}_{\leq t-1})}\left[\mathrm{KL}\left[q_{\boldsymbol{\psi}}(\mathbf{z}_t|\mathbf{z}_{t-1},\mathbf{a}_{t-1},\mathbf{o}_t) \,\|\, p_{\boldsymbol{\theta}}(\mathbf{z}_t|\mathbf{z}_{t-1},\mathbf{a}_{t-1})\right]\right]. \quad (3)$$

If one instead uses the same factorization assumptions as the generative model, the inference distribution factorizes as $q_{\boldsymbol{\psi}}(\mathbf{z}_{\leq T}|\mathbf{o}_{\leq T},\mathbf{a}_{\leq T}) = q_{\boldsymbol{\psi}}(\mathbf{z}_0|\mathbf{o}_{\leq T},\mathbf{a}_{\leq T})\prod_{t=1}^{T} q_{\boldsymbol{\psi}}(\mathbf{z}_t|\mathbf{z}_{t-1},\mathbf{a}_{\geq t-1},\mathbf{o}_{\geq t})$. Inserting this factorization into the general lower bound leads to $\mathcal{L}_{\mathrm{ssm}}(\mathbf{o}_{\leq T}, \mathbf{a}_{\leq T}) =$

$$\sum_{t=1}^{T} \mathbb{E}_{q_{\boldsymbol{\psi}}(\mathbf{z}_t|\mathbf{o}_{\leq T},\mathbf{a}_{\leq T})}\left[\log p_{\boldsymbol{\theta}}(\mathbf{o}_t|\mathbf{z}_t)\right] - \mathbb{E}_{q_{\boldsymbol{\psi}}(\mathbf{z}_{t-1}|\mathbf{o}_{\leq T},\mathbf{a}_{\leq T})}\left[\mathrm{KL}\left[q_{\boldsymbol{\psi}}(\mathbf{z}_t|\mathbf{z}_{t-1},\mathbf{a}_{\geq t-1},\mathbf{o}_{\geq t}) \,\|\, p_{\boldsymbol{\theta}}(\mathbf{z}_t|\mathbf{z}_{t-1},\mathbf{a}_{t-1})\right]\right]. \quad (4)$$

Opposed to the *RSSM*-objective, this objective uses the smoothed belief-state $q_{\boldsymbol{\psi}}(\mathbf{z}_t|\mathbf{o}_{\leq T},\mathbf{a}_{\leq T})$ and the smoothed dynamics $q_{\boldsymbol{\psi}}(\mathbf{z}_t|\mathbf{z}_{t-1},\mathbf{a}_{\geq t-1},\mathbf{o}_{\geq t})$. The variational distribution considers all future information until $t = T$ and thus can theoretically yield a tight bound.

These differences have interesting effects on the learned transition model $p_{\boldsymbol{\theta}}(\mathbf{z}_t|\mathbf{z}_{t-1},\mathbf{a}_{t-1})$. For the *RSSM*, the belief over past states can, by definition, not change due to additional observations. Thus, any discrepancy between this past belief and the current observation must be explained by the transition model. In contrast, a smoothing inference can also explain the discrepancy by propagating information from observations to past beliefs. In Equation 3, this observation is reflected in the expected KL-term, where the expectation does not consider $\mathbf{o}_t$ or other future observations. Thus, the transition model has to explain the transitions from a given $\mathbf{z}_{t-1}$ to $\mathbf{z}_t$, even if $\mathbf{z}_{t-1}$ would be rendered implausible by a future observation.

For a thought experiment illustrating these effects, consider the following scenario. You meet a person holding a box, and they tell you there is a hamster inside. As your prior experience is that people are

usually trustworthy, you chose to believe them. Next, the box opens, and a cat jumps out. As you trust your eyes, you now believe it is a cat. Yet, under the *RSSM*-assumptions, you cannot revise your belief of the first time step and thus still believe it originally was a hamster. When updating your model based on this interaction, you would learn that hamsters can turn into cats, as you cannot capture the arguably more likely explanation that the person lied. Learning under these assumptions requires you to model unlikely events as more likely than they are. Thus, you overestimate the aleatoric uncertainty in the world.

More formally, we can demonstrate this effect using a simple linear-Gaussian State Space Model without actions. We will use a state dimension of 4 and the ground-truth generative model given by

$$p(\mathbf{z}_0) = \mathcal{N}(\mathbf{z}_0|\mathbf{0},\mathbf{I}), \quad p(\mathbf{o}_t|\mathbf{z}_t) = \mathcal{N}(\mathbf{o}_t|\mathbf{I}\mathbf{z}_t, 0.025\mathbf{I}), \quad p(\mathbf{z}_{t+1}|\mathbf{z}_t) = \mathcal{N}(\mathbf{z}_{t+1}|\mathbf{A}\mathbf{z}_t, 0.01\mathbf{I})$$

where $\mathbf{I}$ denotes the identity matrix. The transition matrix $\mathbf{A}$ induces a slightly damped, oscillating behavior. The complete matrix $\mathbf{A}$, together with further details regarding the exact setup of this experiment, can be found in Appendix B. Using this generative model, we generate $1,000$ sequences of length 50. Even in this simple setting, computing the optimal inference distribution for the *RSSM*-bound (Equation 3) is impossible, and we thus resort to numerical methods. We parameterize $q_{\boldsymbol{\psi}}(\mathbf{z}_t|\mathbf{z}_{t-1}, \mathbf{o}_t)$ as locally linear-Gaussian distributions and learn the parameters using a neural network. For the SSM-bound (Equation 4 we can either compute the optimal inference in closed form, or again parameterize $q_{\boldsymbol{\psi}}(\mathbf{z}_t|\mathbf{z}_{t-1}, \mathbf{o}_{\geq t})$ as a neural network. To condition on future observations, we use a GRU (Cho et al., 2014) which runs backward over the observation sequence. For the generative model, we learn a transition matrix $\tilde{\mathbf{A}}$ and an isotropic covariance $\tilde{\sigma}\mathbf{I}$ jointly with the parameters of the inference model, i.e., $\boldsymbol{\theta} = \{\tilde{\mathbf{A}}, \tilde{\sigma}\}$. We fix the remaining parts of the generative model to the ground-truth values. Figure 2 summarizes the results and demonstrates that the *RSSM*-bound leads to a suboptimal inference and consequently to learning a wrong model. In particular, we can see that for the *RSSM*, the transition variance $\tilde{\sigma}$ is much larger than the ground-truth value $\sigma = 0.01$ as all unexpected observations have to be explained by the transitions instead of correcting the beliefs of past time steps.

## 2.3 The Interplay of Policy Optimization and Regularization

Despite the theoretical considerations in the previous section, *RSSMs* work well for model-based RL. It is well known that model-based RL suffers from an *objective mismatch* (Luo et al., 2019; Lambert et al., 2020) issue. This issue arises because the model aims to maximize the ELBO (Equation 1) but is evaluated based on the agent's reward. The effect is further amplified by the distribution shift between training and data collection, as data collection is typically performed only after a policy improvement step. In RL, we explicitly want the agent to explore unseen parts of the state-action space, encountering observations the model has not seen before. Thus, training the underlying model requires careful regularization such that wrong predictions do not prevent the agent from exploring relevant parts of the state space. Many model-based RL approaches (Chua et al., 2018; Janner et al., 2019) handle this issue by explicitly modeling the epistemic uncertainty of the model, which is not required by the *RSSM*. Instead, we argue that *RSSMs* rely on the overestimated aleatoric uncertainty caused by suboptimal inference to address *objective mismatch* in a more heuristic manner. The overestimation implicitly regularizes the *RSSM* as it forces the transition model to model unlikely events with higher probability. This regularization thus implicitly prevents overconfident model predictions due to overfitting. Yet, while it alleviates the *objective mismatch* issue, there are also drawbacks to this heuristic solution. First, it complicates the model design, analysis, and improvement of *RSSMs*. As already observed by Hafner et al. (2019), a fully stochastic model based on the *RSSM*-assumptions underperforms without additional measures as it fails at reliably propagating information for multiple time steps. As a remedy, Hafner et al. (2019) introduce a *deterministic-path*, i.e., a Gated Recurrent Unit (Cho et al., 2014), and base the belief update on this instead of the stochastic belief. Second, as we show in Section 4.2, there are settings where appropriately capturing the aleatoric uncertainty is important, and failing to do so can hurt performance.

In a first attempt to address those issues, we minimally adapt the *RSSM* to be capable of smoothing. This Smoothing *RSSM* uses a GRU which is added before the actual *RSSM* and runs backwards over the representations extracted by the encoder, effectively accumulating all future information. The *RSSM* then receives the output of this GRU instead of the original observation as input. This model is an extension of the

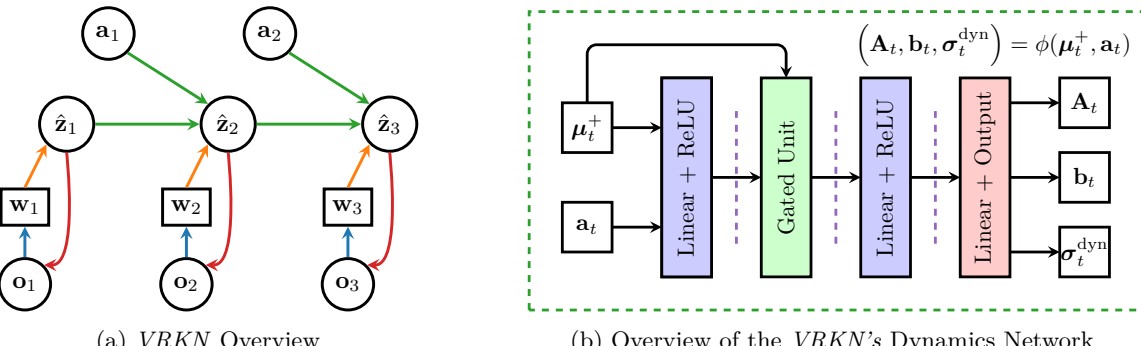

(a) *VRKN* Overview

(b) Overview of the *VRKN's* Dynamics Network

Figure 3: **(a)** We use an encoder (blue) to extracts an intermediate representation $\mathbf{w}_t$ and the observation noise $\boldsymbol{\sigma}_t^{\mathbf{w}}$ from the original observation $\mathbf{o}_t$. Both, $\mathbf{w}_t$ and $\boldsymbol{\sigma}_t^{\mathbf{w}}$, are then used to update the state estimate $\hat{\mathbf{z}}_t$ using the closed-form Kalman belief update (Bayes rule for Gaussians) (orange). A decoder (red) reconstructs the observation given a state sample. We propagate the latent state estimate to the next time step using the transition model $p_{\boldsymbol{\theta}}(\mathbf{z}_{t-1}|\mathbf{z}_t, \mathbf{a}_{t-1})$ (green). The parameters of this distribution, $\mathbf{A}_t$, $\mathbf{b}_t$, and $\boldsymbol{\sigma}_t^{\mathrm{dyn}}$, are computed using the network $\phi(\boldsymbol{\mu}_t^+, \mathbf{a}_t)$, shown in **(b)**. The network receives the current posterior mean $\boldsymbol{\mu}_t^+$ and the action $\mathbf{a}_t$ as inputs. Thus, the resulting dynamics are linearized around the posterior mean, which allows closed-form belief propagation, similar to an extended Kalman Filter. For stability during training, the network includes a gated unit, i.e., a GRU cell, which receives the posterior mean $\boldsymbol{\mu}_t^+$ at the memory input. For a Bayesian treatment of the transition model's parameters, we include Monte Carlo Dropout layers at the positions indicated by the purple dashed lines.

neural network based inference model for $\mathcal{L}_{\mathrm{ssm}}$, introduced in the previous section ( $\mathcal{L}_{\mathrm{ssm}}(\mathrm{CF})$ in Figure 1). We compare this approach to the original *RSSM* and present the results in Section 4.1, in particular Figure 4. Those results show that the performance decreases compared to the original *RSSM*. We argue that with a proper inference, the overestimated aleatoric uncertainty no longer regularizes the model, making it more prone to *objective mismatch*. Following other methods, we try to improve the results by modeling epistemic uncertainty. To this end, we use Monte Carlo Dropout (MCD) but find that it does not help to improve the Smoothing *RSSM's* performance. From these results, again presented in Figure 4, we conclude that solely addressing the suboptimal inference assumption is insufficient, but we also need to rethink the model's parameterization. We postulate that the additional GRU for the backward pass is a poor inductive bias and that we require a smoothing approach that adds as little complexity as possible to the model. In the next section, we will introduce an architecture that allows for parameter-free smoothing.

## 3 Variational Recurrent Kalman Networks

To provide a theoretically better-grounded alternative to the *RSSM*, we require a model which allows tractable inference while still scaling to complex image-based control tasks. Further, the architecture should allow efficient computation of smoothed and posterior state beliefs and dynamics. While we need smoothed beliefs for training, we require (filtered) posteriors for online control. In particular, as only the smoothed beliefs are explicitly used for training, the model needs to provide a strong inductive bias that enables it to still produce reasonable posterior estimates.

To meet these criteria, we introduce a new parametrization of the latent dynamics based on a linear-Gaussian state space model (LGSSM) (Murphy, 2012) embedded in a latent space. The linear-Gaussian assumptions allow a rigorous treatment of uncertainties while working in a learned latent space allows for modeling high-dimensional and non-linear systems. Inference in this LGSSM amounts to (extended) Kalman filtering and smoothing, enabling smoothing inference without introducing additional parameters besides the latent observation and dynamics model. Due to this property, the architecture can perform proper smoothing inference for training and also produce reasonable posterior beliefs for online control. As an additional benefit, the formulation can naturally handle missing observations and the fusion of multiple sensors, making it amenable to many realistic problems. To include epistemic uncertainty in our approach, we use Monte

Carlo Dropout (Gal & Ghahramani, 2016) for a Bayesian treatment of the LGSSM's transition model. We name the resulting approach *Variational Recurrent Kalman Network (VRKN)*. Figure 3 shows a schematic overview.

In the following, we first introduce the *VRKN*'s dynamics model, which is shared between the inference and generative parts of the model. Next, we introduce the parameterization of the inference model $q_{\boldsymbol{\psi}}$ and the generative model $p_{\boldsymbol{\theta}}$. We describe how to train the model and use it for control and conclude by elaborating on the natural fusion mechanism.

### 3.1 The *VRKN*'s Dynamics Model

Both, the *VRKN*'s inference model $q_{\boldsymbol{\psi}}$ and the generative model $p_{\boldsymbol{\theta}}$, use the same dynamics model. Such parameter sharing is common for variational time-series models (Karl et al., 2016; Fraccaro et al., 2017; Hafner et al., 2019; Klushyn et al., 2021), as it simplifies the architecture, reduces the number of parameters, and simplifies training.

We model the latent dynamics as

$$p_{\boldsymbol{\theta}}(\mathbf{z}_{t+1}|\mathbf{z}_t, \mathbf{a}_t) = \mathcal{N}\left(\mathbf{z}_{t+1}|\mathbf{A}_t(\boldsymbol{\mu}_t^+, \mathbf{a}_t)\mathbf{z}_t + \mathbf{b}_t(\boldsymbol{\mu}_t^+, \mathbf{a}_t), \boldsymbol{\sigma}_t^{\mathrm{dyn}}(\boldsymbol{\mu}_t^+, \mathbf{a}_t)\right),$$

where $\boldsymbol{\mu}_t^+$ denotes the mean of the posterior state estimate $q_{\boldsymbol{\psi}}(\mathbf{z_t}|\mathbf{o}_{\leq t}, \mathbf{a}_{\leq t})$ provided by the inference model. Thus, we learn a linearized dynamics around the current posterior mean. This parametrization builds on the work of Shaj et al. (2020), who propose using a model that is locally linear in the state while depending non-linearly on the action. Building on the assumption that there is no uncertainty in the actions, it is maximally flexible while still allowing the usage of extended Kalman filtering (Jazwinski, 1970). Here, the linearization around the current posterior mean $\boldsymbol{\mu}_t^+$ enables closed-form propagation of state beliefs.

We model $\mathbf{A}_t$ to be a diagonal matrix which is emitted together with the offset term $\mathbf{b}_t$ and the transition noise $\boldsymbol{\sigma}_t^{\mathrm{dyn}}$ by a single neural network $\left(\mathbf{A}_t, \mathbf{b}_t, \boldsymbol{\sigma}_t^{\mathrm{dyn}}\right) = \phi(\boldsymbol{\mu}_t^+, \mathbf{a}_t)$. We carefully design this network to prevent the state estimates and gradients from growing indefinitely during training. First, as $\mathbf{A}_t$ is diagonal, its values are its eigenvalues, and constraining them in an appropriate range ensures stable dynamics. To this end, we use an activation of the form $f(x) = s \cdot \mathrm{sigmoid}(x+b) + m$ where we choose $s$, $b$, and $m$ such that it saturates at 0.1 and 0.99 while $f(0) = 0.9$. Here, the intuition is that we want plausible and stable dynamics, which we initialize as a slightly dampened system. Second, we employ a gating mechanism to mitigate problems with vanishing and exploding gradients. To this end, we use a standard GRU cell implementation but feed the posterior mean $\boldsymbol{\mu}_t^+$ into the memory input, i.e., $\phi(\boldsymbol{\mu}_t^+, \mathbf{a}_t) = \phi_2(\mathrm{GRU}(\phi_1(\boldsymbol{\mu}_t^+, \mathbf{a}_t), \boldsymbol{\mu}_t^+))$. The resulting model is still fully stochastic and linear in $\mathbf{z}_t$. In contrast to the *RSSM*, it does not use a *determinstic-path* as the GRU cell does not introduce an additional deterministic memory and is used solely for addressing problems arising from unstable dynamics and gradients. We empirically found this helpful and took inspiration from standard recurrent architectures (Hochreiter & Schmidhuber, 1997; Cho et al., 2014), where gating mechanisms are a well-known approach to mitigate instabilities. Figure 3b provides an overview of the dynamics model architecture.

**Modeling Epistemic Uncertainty.** As discussed in Section 2.3, the overestimated aleatoric uncertainty of the *RSSM*'s transition model avoids overconfident estimates due to overfitting and compensates for the lack of explicit epistemic uncertainty. Thus, as our approach captures the aleatoric uncertainty correctly, we need to explicitly consider epistemic uncertainty to obtain a model that is also useable for policy optimization. We use Monte Carlo Dropout (Gal & Ghahramani, 2016) due to its simplicity and include corresponding layers at appropriate points in the transition model. Those point are shown in Figure 3b.

### 3.2 Inference Model

We aim for a model that can work with high-dimensional observations which depend non-linearly on the latent state while still allowing efficient inference of filtered and smoothed beliefs. Thus, we introduce an auxiliary, intermediate representation $\mathbf{w}_t$ for the original observations $\mathbf{o}_t$. Such an intermediate representation allows

us to capture the highly complex relations between state and observations in the mapping from $\mathbf{o}_t$ to $\mathbf{w}_t$ while using a simple observation model $q_\psi(\mathbf{w}_t|\mathbf{z}_t)$ in the latent space. This approach is common for variational latent state space models (Karl et al., 2016; Fraccaro et al., 2017; Klushyn et al., 2021) as it enables efficient inference in latent space. These approaches model $\mathbf{w}_t$ as a latent Gaussian random variable and learn it's mean and variance based on $\mathbf{o}_t$. They then sample $\mathbf{w}_t$ given these parameters. Yet, this assumption complicates inference and training, as now an additional latent variable, $\mathbf{w}_t$, has to be addressed. Further, it makes uncertainty propagation from observations to states harder (Becker et al., 2019; Volpp et al., 2021), as explicit uncertainty information is lost during sampling. Thus, following (Haarnoja et al., 2016; Becker et al., 2019; Shaj et al., 2020; Volpp et al., 2021), we instead formulate the observation model as $q_\psi(\mathbf{w}_t|\mathbf{z}_t) = \mathcal{N}\left(\mathbf{w}_t|\mathbf{z}_t, \boldsymbol{\sigma}_t^{\mathbf{w}}\right)$, where $\mathbf{w}_t = \text{enc}_o(\mathbf{o}_t)$ and the diagonal covariance $\boldsymbol{\sigma}^{\mathbf{w}_t} = \text{enc}_\sigma(\mathbf{o}_t)$ are given by an encoder network taking the original observation. In practice, the encoder consists of a single neural network with two output heads. This formulation does not suffer from the previously mentioned issues. First, as it models $\mathbf{w}_t$ as a direct mapping of the observation and not a random variable, we do not need to infer $\mathbf{w}_t$ but can assume it is observable. This assumption results in a simplified inference and training objective as we do not have to account for unobserved latent variables other than the latent state itself. Second, the encoder can directly extract the uncertainty in the observation, which tells the inference procedure how much it can rely on $\mathbf{w}_t$. For example, when estimating $\mathbf{w}_t$ from images, some images might contain certain information, e.g., the positions of an object, while others do not. The former case would result in a low uncertainty and the latter in a high one. Given $\mathbf{w_t}$ and $\boldsymbol{\sigma}_t^{\mathbf{w}}$ we can update belief states using the Kalman update, i.e., Bayes rule for Gaussian distributions,

$$q_\psi(\mathbf{z}_t|\mathbf{o}_{\leq t}, \mathbf{a}_{\leq t-1}) = \frac{1}{Z} q_\psi(\text{enc}_o(\mathbf{o}_t)|\mathbf{z}_t) q_\psi(\mathbf{z}_t|\mathbf{o}_{\leq t-1}, \mathbf{a}_{\leq t-1}).$$

Here, $\boldsymbol{\sigma}_t^{\mathbf{w}}$ contributes to the computation of the Kalman gain, which highlights how the model uses the encoder's uncertainty to trade-off information from the prior belief and observation

The inference observation model is combined with the shared locally linear dynamics model $q_\psi(\mathbf{z}_{t+1}|\mathbf{z}_t, \mathbf{a}_t) = p_{\boldsymbol{\theta}}(\mathbf{z}_{t+1}|\mathbf{z}_t, \mathbf{a}_t)$ to obtain a latent LGSSM for inference. Given this dynamics model, we can forward beliefs in time using closed-form Gaussian marginalization,

$$q_\psi(\mathbf{z}_{t+1}|\mathbf{o}_{\leq t}, \mathbf{a}_{\leq t}) = \int q_\psi(\mathbf{z}_{t+1}|\mathbf{z}_t, \mathbf{a}_t) q_\psi(\mathbf{z}_t|\mathbf{o}_{\leq t}, \mathbf{a}_{\leq t-1}) d\mathbf{z}_t.$$

Repeating the described forms of forwarding beliefs in time and updating them based on observations amounts to standard (extended) Kalman filtering (Kalman, 1960; Jazwinski, 1970). Similarly, given these models, we can compute smoothed belief states using the Rauch-Tung-Striebel (Rauch et al., 1965) equations. In particular, the smoothing only requires beliefs and transition models computed during filtering and does not introduce additional learnable parameters. Note that both the filtering and smoothing equations simplify under our assumptions of diagonal covariances and are thus efficient and scalable. Thus, the smoothing inference only slightly increases the computational load during training. Especially in the image-based settings considered here, the computational cost largely stems from encoding and reconstructing observations.

### 3.3 Generative Model

Recall that we assume a generative State Space Model consisting of three parts. Those are the dynamics model, the observation model, and the initial state distribution. First, we have the dynamics model $p_{\boldsymbol{\theta}}(\mathbf{z}_{t+1}|\mathbf{z}_t, \mathbf{a}_t)$ as described in Section 3.1. Second, we assume a Gaussian generative observation model $p_{\boldsymbol{\theta}}(\mathbf{o}_t|\mathbf{z}_t)$ parameterized by a neural network (decoder). Following Hafner et al. (2019), we parameterize the mean by a neural network and assume a fixed variance. This parameterization is often empirically beneficial, especially in the image-based setting considered in our experiments, but not a theoretical constraint. We could also parameterize the variance by the network if required. Third, we have an initial state distribution $p_{\boldsymbol{\theta}}(\mathbf{z}_0)$. Here we use a Gaussian with zero mean and a learned diagonal variance which we initialize with the identity matrix.

### 3.4 Stochastic Gradient Variational Bayes for Model Learning

We train our model using a version of stochastic gradient variational Bayes (Kingma & Welling, 2013) and maximize $\mathcal{L}_{\text{ssm}}$, (Equation 4). Like most approaches building on (Kingma & Welling, 2013) we approximate all expectations in Equation 4 using Monte Carlo estimation with a single sample from the latent variable and jointly optimize the parameters of $q_{\psi}$ and $p_{\theta}$. For training, we infer the required belief states using the Kalman-based smoothing procedure described in Section 3.2. To compute the KL term in Equation 4 we additionally need the smoothed dynamics $q_{\psi}(\mathbf{z}_{t+1}|\mathbf{z}_t, \mathbf{a}_{\geq t}, \mathbf{o}_{\geq t+1})$ which we obtain with minimal overhead by extending the RTS equations as detailed in subsection A.2.

The *VRKN's* formulation tightly couples posterior and smoothed beliefs by a fixed (not learned), deterministic, and well-motivated procedure. Smoothing using the Rauch-Tung-Striebel equations ensures that reasonable smoothed beliefs can only stem from reasonable posterior beliefs. Thus, while the posterior belief is not explicitly part of the objective, it is still learned during training as a prerequisite of the smoothed beliefs.

### 3.5 Using the Model for Online Control and Reinforcement Learning

When using the model for online control, we cannot smooth but have to act based on samples from the posterior belief $q_{\psi}(\mathbf{z}_t|\mathbf{a}_{\leq t}, \mathbf{o}_{\leq t})$. We can rely on Kalman filtering, as described in Section 3.2, to obtain this posterior belief and omit the smoothing step, as future observations are unavailable. For the *VRKN*, the tight coupling between smoothed and posterior beliefs ensures the posteriors are reasonable and usable for control.

To act based on the latent state beliefs, we add a decoder network to predict rewards from latent states. Following (Hafner et al., 2019), this decoder is trained by adding a reconstruction loss term to the objective. Given the predicted reward, various forms of control are applicable to act optimally. Yet, in this work, we focus on the underlying state space model and thus reuse two previously introduced methods building on the *RSSM* (Hafner et al., 2019; 2020). The *PlaNet* (Hafner et al., 2019) approach plans actions using the cross entropy method by rolling out trajectories on the model. The *Dreamer* (Hafner et al., 2020) approach learns a parametric policy and value function based on latent imagination. Such latent imagination uses the model as a differentiable simulator and optimizes the policy based on the estimated values of predicted future states.

### 3.6 Sensor Fusion

Given the possibility of using the Kalman update for incorporating observations, we can use the *VRKN* for a simple but principled approach to sensor fusion. Formally, we assume the observation $\mathbf{o}_t$ factorizes into $K$ different observations $\mathbf{o}_t^{(k)}$, i.e., $p_{\theta}(\mathbf{o}_t|\mathbf{z}_t) = \prod_{k=1}^{K} p_{\theta}(\mathbf{o}_t^{(k)}|\mathbf{z}_t)$. Those observations can be of various modalities and be available at different frequencies, e.g., high-frequency velocity information from an inertial measurement unit and low-frequency camera images of the surroundings. In this scenario, we have $K$ encoders and $K$ decoders, one for each $\mathbf{o}_t^{(k)}$. Similar to more traditional sensor fusion approaches (Gustafsson, 2010), we then accumulate the intermediate observation representation $\mathbf{w}_t^{(k)}$ by repeatedly applying the Kalman update. This approach reflects the invariance to permutations of all observations for a single time step. Furthermore, it enables the model to omit the update if some of the $K$ observations are unavailable for a time step.

## 4 Evaluation

We compare the *VRKN*, the original *RSSM*, and the modified *RSSM* version introduced in Section 2.3 on image-based continuous control tasks using the DeepMind Control Suite (Tassa et al., 2020). Prior works (Lambert et al., 2020; Lutter et al., 2021) concluded that the model's predictive performance is often uninformative about the quality of the model-based agent. We concluded the same after preliminary experiments and want to study the effects of the different assumptions and parametrizations on the performance in a model-based RL setting. Thus, we evaluate the state space models as backbones for model-based

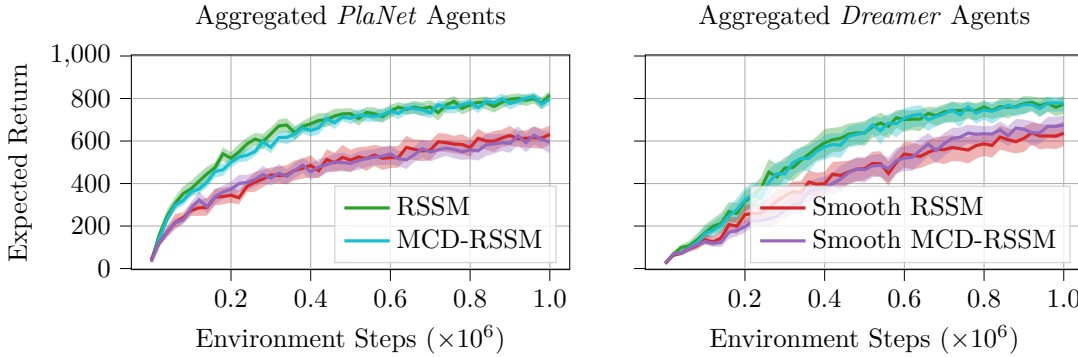

Figure 4: Comparison of the *RSSM*, the simple smoothing extension (Smooth *RSSM*) as well as versions of both models using Monte Carlo Dropout (MCD) to capture epistemic uncertainty in the dynamics (MCD-*RSSM* and Smooth MCD-*RSSM*). Note that we use different tasks for *PlaNet* and *Dreamer*-based agents. Thus the left and right plots are not directly comparable. For both types of agents, we find that proper inference by smoothing deteriorates performance. The additional epistemic uncertainty does not compensate for this decrease in performance.

agents and directly consider the achieved reward. As mentioned, we use both the *PlaNet* and the *Dreamer* approaches for control and thus closely follow the experiment setup described by Hafner et al. (2019; 2020). Appendix C gives further details about the experimental setup and used baselines.

For our experiments, we run a varying number of tasks from the DeepMind Control Suite (Tassa et al., 2020) and report aggregated results over these tasks. Such aggregation is possible as the rewards in the DeepMind Control Suite are normalized, and all sequences are of equal length (1,000 steps). Thus the returns are normalized, and aggregation yields better performance estimates (Agarwal et al., 2021). Again following the suggestions of Agarwal et al. (2021), we report interquartile means while indicating 95% stratified bootstrap confidence intervals by shaded areas. These metrics are computed using the provided library[3]. We base our conclusions on those aggregated results. Unless noted otherwise, we use 10 seeds for each agent-environment pair, train for 1 million environment steps, and approximate the expected return using 10 rollouts. Appendix D provides reward curves for the individual tasks and additional quantitative results, such as box plots of their final performance.

## 4.1 Evaluation of the Effect of Epistemic Uncertainty on Different Smoothing Architectures

We start our evaluation by comparing the original *RSSM* with its extended smoothing version using a GRU with and without Monte Carlo Dropout (MCD), described in Section 2.3. For completeness, we also include a version of the original-*RSSM* with MCD to model epistemic uncertainty. For the *PlaNet*-agents, we evaluate the 6 tasks originally used in (Hafner et al., 2019). For the *Dreamer*-agents we use 8 tasks. Those are Cheetah Run, Walker Walk, Cartpole Swingup, Cup Catch, Reacher Easy, Hopper Hop, Pendulum Swingup, and Walker Run. As indicated by the results in Figure 4, the proper smoothing inference significantly deteriorates performance. Adding epistemic uncertainty in the form of MCD does, on average, neither affect the performance of the original *RSSM* nor the smoothing *RSSM*. These results underpin the argument that naive smoothing with the *RSSM* gives suboptimal performance. Additionally, they show that epistemic uncertainty in the form of Monte Carlo Dropout cannot serve as a remedy for the *RSSM*.

Next, we compare the *VRKN* to the *RSSM* using the same environments and also include a version of the *VRKN* without Monte Carlo Dropout (MCD), i.e., without epistemic uncertainty, dubbed *VRKN (no MCD)*. Figure 5 shows the aggregated results for both *PlaNet* and *Dreamer* agents. Those results demonstrate that matching the *RSSM's* performance is possible with a principled smoothing inference by explicitly modeling epistemic uncertainty and providing a more appropriate inductive bias. Additionally, we find the *VRKN* tends to reach a given performance with fewer environment interactions, especially for the *Dreamer*-agents.

---

[3]rliable: `https://github.com/google-research/rliable`

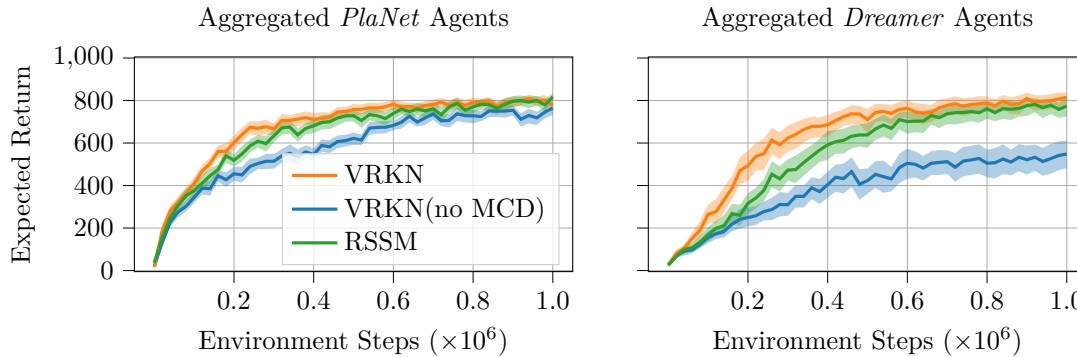

Figure 5: Comparison of *VRKN* and *RSSM* based *PlaNet* and *Dreamer* agents on standard DeepMind Control Suite tasks. Note that we use different tasks for *PlaNet* and *Dreamer*-based agents. Thus the left and right plots are not directly comparable. The *VRKN* without epistemic uncertainty (*VRKN (no MCD)*) cannot compete with the *RSSM*-agents, especially using the more involved *Dreamer*-based agents. Yet, the *VRKN* profits from the epistemic uncertainty, and closes the performance gap to the *RSSM* and even shows a tendency to be more sample efficient.

Together, the results from Figure 4 and Figure 5 emphasize the importance of regularization for model-based RL. This regularization can be either implicitly by suboptimal inference or explicitly by capturing epistemic uncertainty. Additionally, they indicate that epistemic uncertainty alone is insufficient for approaches using a correct smoothing inference. Yet, the *VRKN's* appropriate inductive bias makes the more principled approach of combining proper inference with explicit epistemic uncertainty feasible.

### 4.2 Evaluation on Tasks where Aleatoric Uncertainty Matters

The standard versions of the Deep Mind Control Suite Tasks have a deterministic simulation and rendering process. Thus, their aleatoric uncertainty is low. To better analyze the approaches' capabilities to capture and handle aleatoric uncertainty, we design tasks where doing so is necessary. To this end, we modify the Cheetah Run, Walker Walk, Cartpole Swingup, and Cup Catch tasks. First, we introduce transition noise by adding Gaussian noise to the actions before execution. We added noise with a standard deviation of 0.2 for Cheetah Run and Walker Walk and 0.3 for Cartpole Swingup and Cup Catch. Note that valid actions are between $-1$ and $1$ for all tasks. Second, we modify the observations to contain only partial information, are missing for several time steps, or are available in different modalities at different frequencies. For details, we refer to the individual experiments below.

We use these tasks to study the effects of appropriately capturing aleatoric uncertainty in tasks where this form of uncertainty matters. We compare *VRKN*-based and the *RSSM*-based *Dreamer* agents and find that precise estimation of the aleatoric uncertainty makes a difference. Contrary to the original tasks, where *RSSM* and *VRKN*-based agents performed similarly, agents building on the *VRKN* outperform their *RSSM*-based counterparts in all considered scenarios.

#### 4.2.1 Partial Observability through Occlusions

In a first approach to include observation uncertainty, we render two types of occlusions over the images, i.e., discs and walls. See Figure 6 for an explanation and some examples. Due to these occlusions, the individual observations have varying amounts of relevant information. Thus, the models need to correctly capture uncertainties in the system, allowing them to trade off information from the prior belief and current observation. For training, we use masked reconstruction, i.e., only non-occluded pixels contribute to the reconstruction loss[4]. We also consider a baseline where we train the model solely to reconstruct the reward to show that the approaches can still extract information from the highly occluded observations. Figure 7

---

[4]We want to emphasize that we do not consider the availability of such loss masks a realistic scenario but see the task as a reasonable benchmark to evaluate the models' capabilities to cope with uncertainties.

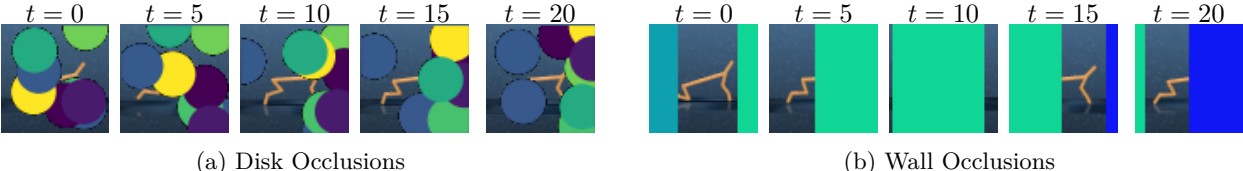

(a) Disk Occlusions          (b) Wall Occlusions

Figure 6: We introduce two types of occlusions to induce partial observability. **(a) Disk Occlusion:** Slow-moving disks float through the image and bounce off its walls. **(b) Wall Occlusions:** Walls slide over the image from right to left at a constant speed. Their width is sampled randomly between half the image width and the image width. In both cases, the models have to accurately estimate which parts of the system are visible and which are not. Yet, the wall-based occlusions are more correlated over time and require memorization and prediction over longer time periods.

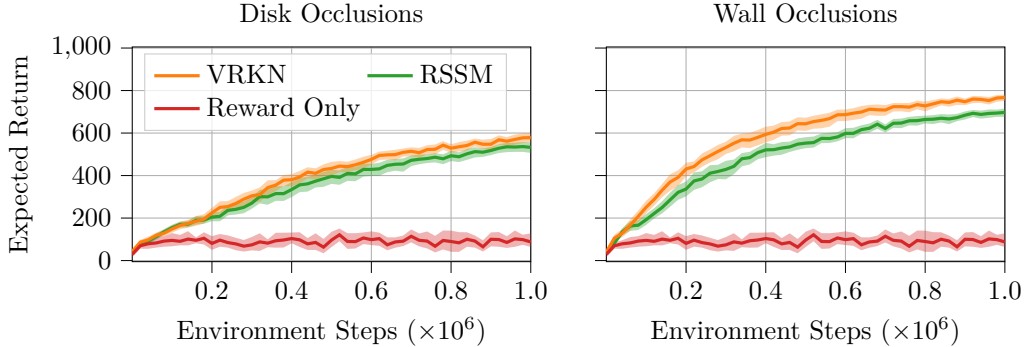

Figure 7: Aggregated results for the partial observability experiments with both occlusion types. For the experiments, except for the reward-only baseline, we used 20 seeds per task. Both approaches clearly perform better than the reward-only baseline, showing that they can extract useful information from the occluded images. Yet, especially in the wall-occlusion task, the *VRKN*-based agents outperform the *RSSM*-based agents. They do not only achieve a higher reward but also converge faster. Especially in the wall occlusion task, it is insufficient to only extract information from the occluded images, but a reasonable belief also needs to be sustained over time.

shows the results of the comparison. Under the heavy occlusions considered here, the *VRKN* performs significantly better than the *RSSM*. In particular for the wall occlusions, the *VRKN*-based agents achieve a higher reward using fewer environment interactions. The wall occlusions are more correlated, i.e., occluded parts in one image are more likely also occluded in the previous and next images. Thus, they test the approaches' capability to not only form a reasonable belief but also to propagate it for multiple time steps.

Additionally, we qualitatively compare images reconstructed from posterior beliefs of both approaches to gain further insights into the quality of the belief state. Figure 8 shows some of those reconstructions for the Cup Catch task, further images can be found in Appendix D.6. From these images, it appears that the *VRKN* better captures the actual system state and uncertainty and thus allows the model-based agent to achieve a higher reward. Yet, while the reconstructions are visually much better, we only observe a small increase in performance, which we attribute to *objective missmatch*.

### 4.2.2 Dealing with Missing Observations

Another common setting where estimating aleatoric uncertainty is important is missing observations. It requires the models to propagate a reasonable belief without information and update it if an observation is available. Additionally, if no observations are available for multiple time steps, the belief state gradually drifts away from the real state due to noise and model inaccuracies. Once a new observation arrives, the discrepancy between this observation and the belief has to be explained. Recall that, due to its inference assumptions, the *RSSM* must explain the discrepancy by the last transition and thus will overestimate the

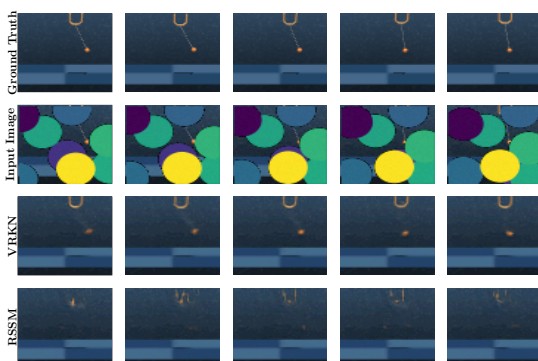

Figure 8: Exemplary sub-sequence of reconstructions, based on the model's posterior beliefs. The first row is the noise-free ground truth image, which the models never see. The second row is the model input, followed by the *VRKN* and the *RSSM* reconstructions. Even though the ball is partly visible in most images, the *RSSM* fails to reconstruct its position. The *VRKN* manages to do so and even provides a reasonable estimate for cup position. These results indicate the *VRKN's* improved ability to capture the system state in noisy scenarios.

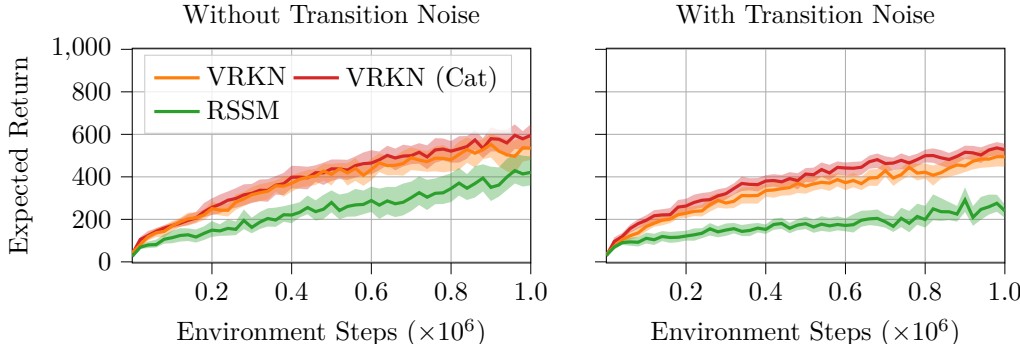

Figure 9: Aggregated results in the missing observations setting with and without transition noise. First, we see that the *VRKN* works equally well if we provide the information about which information are valid explicitly (*VRKN*) or if we provide it by concatenation (*VRKN (Cat)*). Second, the *RSSM* performs significantly worse then both both *VRKN* version. Especially if the system is subject to transition noise it fails to give good results.

aleatoric uncertainty even further. On the other hand, the *VRKN* with its smoothing inference can capture the drift of the belief state and evenly attribute the discrepancy to all transitions since the last observation. Thus the *VRKN* should be better equipped to solve such a task.

We consider a setting where we only provide every $n$-th image, where $n$ is sampled uniformly between 4 and 8. We assume knowledge about whether the observation for a given time step is valid and feed that knowledge to the models as an additional input flag. For valid observation, the model receives the observation and a 1. For invalid observations, it receives a default value and a 0. The *RSSM* handles this flag by appending it to the encoder output[5] or the default value. The *VRKN* allows for two ways of handling the flag. First, we can again concatenate it to the observation before computing the latent observation and corresponding uncertainty estimate (*VRKN (Cat)*). Second, the *VRKNs* design allows us to provide the information more explicitly and the Kalman update step for invalid observations during inference. We evaluate with and without additional transition noise and show the results in Figure 9. Those results underpin the initial hypotheses that the *VRKN* is better equipped to learn in this setting and significantly outperforms the *RSSM*. The improvement is independent of how we provide information about missing information to the *VRKN* and, as expected, more significant with transition noise.

### 4.2.3 Fusing Information from Multiple Sensors at Different Frequencies

We extend the missing observations setting using proprioceptive information which is available at every time step. The exact form of the proprioceptive information is task-dependent. For example, for Cup Catch, we define the cup position as proprioceptive, but not the ball position, which has to be inferred from

---

[5]For the *RSSM* we refer to the convolution network processing the inputs as encoder, see Appendix C.3 for details.

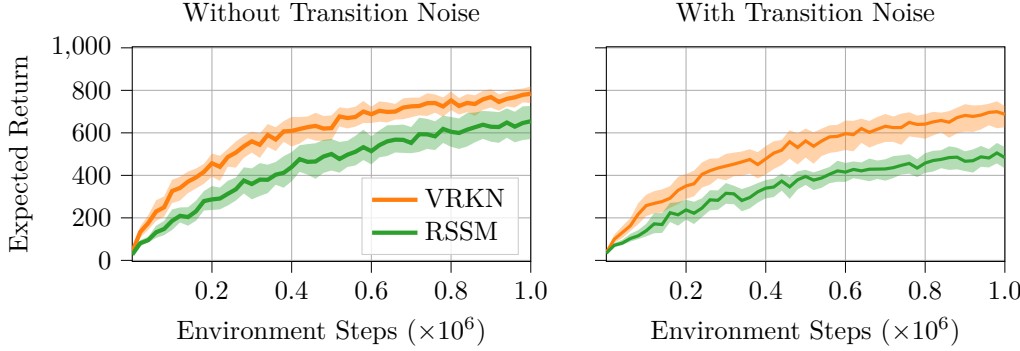

Figure 10: Aggregated results in the fusion setting with and without transition noise. When comparing to Figure 9 we see that both approaches are able to exploit the additional proprioceptive information and improve their performance. Yet, the *VRKN* seems to better exploit and accumulate the available information, as the resulting agents significantly outperform their *RSSM*-based counterparts.

images. Appendix C.4 provides an overview for all considered tasks. This experiment mimics a common robotics scenario where we have proprioceptive information about the robot at high frequencies but need to estimate the environment's state based on lower-frequency images. It tests the approaches' abilities to form reasonable state estimates from observations that arrive in different modalities and at varying frequencies. Here, the models need to trade off information encoded in the prior belief with the information available in both sensor sources. This trade off requires accurate estimates about the aleatoric uncertainty in both state and observations. For the fusion task, the *VRKN* can rely on the natural fusion mechanism described in Section 3. For the *RSSM*, we also use multiple encoder networks, decoders, and loss terms and concatenate the outputs of the encoders before forming the posterior belief. For details, see Appendix C.3.

As shown in Figure 10, the *VRKN* achieves a higher reward than the *RSSM*, especially in the setting with transition noise. This result again emphasizes the *VRKNs* capability to exploit all available information, appropriately capture the system's uncertainty, and form belief states that yield good performance.

## 5  Related Work

***Recurrent State Space Models.*** While earlier works (Wahlström et al., 2015; Watter et al., 2015; Banijamali et al., 2018; Ha & Schmidhuber, 2018; Buesing et al., 2018) discussed and showed the feasibility of control using learned latent state space models, the work originally proposing the *RSSM* (Hafner et al., 2019) was the first to show that such approaches can achieve similar performance to model-free RL on pixel-based complex continuous control tasks while using significantly fewer environment interactions. Since then, Hafner et al. (2020) improved their approach using a parametric policy learned on imagined trajectories and categorical latent spaces (Hafner et al., 2021). These approaches gained interest in the model-based RL community and are empirically successful, yet, little attention has been paid to the underlying state space model itself, the assumptions it builds upon, and its parametrization.

**State Space Models.** The Machine Learning community extensively studied and used state space models (SSMs). Besides classical approaches using linear models (Shumway & Stoffer, 1982) and works using Gaussian Processes (Eleftheriadis et al., 2017; Doerr et al., 2018), most recent methods build on Neural Networks (NNs). The first class of NN-based models of particular relevance for this work embeds linear-Gaussian SSMs (LGSSM) into latent spaces (Watter et al., 2015; Karl et al., 2016; Fraccaro et al., 2017; Banijamali et al., 2018; Becker-Ehmck et al., 2019; Klushyn et al., 2021). These approaches assume actuated systems and learn using stochastic gradient variational Bayes (Kingma & Welling, 2013). Yet, non of these approaches were used to model or even control systems of the complexity considered by (Hafner et al., 2019) and here. They are not directly applicable to these scenarios for various reasons. First, they use full transition matrices and covariances, which prevents them from scaling to sufficiently high-dimensional latent spaces. (Karl et al., 2016; Becker-Ehmck et al., 2019) do not allow smoothing. (Fraccaro et al., 2017;

Klushyn et al., 2021) model the latent observations as random variables which are inferred jointly with the latent states and use constant observation uncertainty for the filtering in the latent space. This choice complicates inference and training and prevents principled usage of the observation uncertainty for filtering. Our parameterization of the LGSSM alleviates these issues by building on factorization assumptions which yield a scalable architecture. Further, it allows smoothing and principled usage of observation uncertainty during filtering by modeling the observations in latent space as deterministic. Finally, non of these approaches considered modeling epistemic uncertainty.

Another class of approaches directly uses NN-based, nonlinear parametrization for SSMs (Archer et al., 2015; Krishnan et al., 2015; Gu et al., 2015; Zheng et al., 2017; Krishnan et al., 2017; Yingzhen & Mandt, 2018; Schmidt & Hofmann, 2018; Naesseth et al., 2018; Moretti et al., 2019). Out of this class, *Structured Inference Networks (SINs)* (Krishnan et al., 2017) are the most relevant for our work. *SINs* build on the same variational objective as *VRKN*, yet without conditioning on actions. The smoothing-*RSSM* baseline introduced in Section 2.3 can be considered an instance of a *SIN*. Yet while it builds on the same loss and fundamental ideas, the underlying NN architecture is different.

**Kalman Updates in Deep Latent Space.** Haarnoja et al. (2016) first proposed using an encoder to extract uncertainty estimates from high-dimensional observations for filtering. They only learned the encoder while assuming the transition dynamics to be known. Becker et al. (2019) proposed an efficient factorization to additionally learn a high-dimensional, latent, locally-linear dynamics model. Shaj et al. (2020) extended this approach by introducing a principled form of action conditioning. While the *VRKN* builds on many of their design choices, there are also considerable differences. Those differences mainly concern the parametrization of the dynamics model and further simplifying the factorization assumptions. These changes are necessary to make the approaches scale to the complex control tasks considered in this work. Additionally, Haarnoja et al. (2016); Becker et al. (2019); Shaj et al. (2020) train using regression and do not learn a full generative model. Thus, they cannot produce the reasonable latent trajectories needed for model-based RL.

**Epistemic Uncertainty for Model-Based RL.** Ample work emphasises the importance of modeling epistemic uncertainty for model-based RL (Deisenroth & Rasmussen, 2011; Chua et al., 2018; Janner et al., 2019) and several authors equipped *RSSMs* with epistemic uncertainty. Okada et al. (2020) use an ensemble of *RSSMs* and showed improved results on modified versions of the Deep Mind Control Suite (Tassa et al., 2020) benchmarks. Sekar et al. (2020) also combine an ensemble with the *RSSM* but focus on exploration and generalization to unseen tasks. Yet, neither of these works questioned the assumptions underlying the *RSSM* or analyzed their effects on the learned models.

# 6 Conclusion

We analyzed the independence assumptions underlying *Recurrent State Space Models (RSSMs)* and found they are theoretically suboptimal. Yet, they implicitly regularize the model by causing an overestimated aleatoric uncertainty and are crucial to the *RSSMs* success in model-based RL. When trying to avoid this heuristic approach and use the correct assumptions while replacing the implicit regularization with a more explicit approach using epistemic uncertainty, we found a simple extension of the *RSSM* architecture is insufficient. Thus, we redesigned the model using well-understood and established components, providing a more appropriate inductive bias for smoothing. The resulting *Variational Recurrent Kalman Network (VRKN)* uses a latent linear Gaussian State Space Model (LGSSM) to address aleatoric uncertainty and Monte-Carlo Dropout to model epistemic uncertainty explicitly. Building on an LGSSM allows exact inference in the latent space using Kalman filtering and smoothing to obtain both smoothed and posterior belief states efficiently. While agents based on the *VRKN* and the *RSSM* perform similar on the standard DeepMind Control Suite (Tassa et al., 2020) benchmarks, the *VRKN*-based agents significantly outperform those using the *RSSM* on tasks where capturing uncertainties is more relevant. Additionally, the *VRKN* provides a natural approach to sensor fusion and outperforms the RSSM on tasks that require fusing sensor observations from several sensors at different frequencies.

**Limitations.** We showed that designing a state space model out of well-founded components that matches or improves the *RSSMs* performance is possible. This insight opens a path to improve them individually.

Yet, here we used simple instances of these components and have not yet further investigated how to improve them. Further, we have not investigated the interplay between the models and the controllers used on top of them but used the control approaches proposed in (Hafner et al., 2019) and (Hafner et al., 2020) with default parameters. Due to the intricate interplay between model learning and using the resulting controller for data collection, it is reasonable to rethink the design of the controller when changing the model.

## Acknowledgments

The authors acknowledge support by the state of Baden-Württemberg through bwHPC, as well as the HoreKa supercomputer funded by the Ministry of Science, Research and the Arts Baden-Württemberg and by the German Federal Ministry of Education and Research.

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
