# OpenReview forum: "On Uncertainty in Deep State Space Models for Model-Based Reinforcement Learning"
_TMLR — Accepted by TMLR_

### Review · Reviewer_TTvU · 2022-08-09

**Summary Of Contributions:**

The paper studies a popular model (RSSM) and its associated learning algorithm used in many recent successful model-based reinforcement learning algorithms. It shows that the model learning algorithm uses a sub-optimal inference scheme leading to an over-estimation of aleatoric uncertainty. The main goal of the paper is to seek a new variational model learning method and a new model that involve an optimal inference scheme.

It empirically shows that, by performing inference in an appropriate way to deal with the over-estimation problem (I can not understand how this is done though), the performance of PlaNet and Dreamer, two popular algorithms using RSSM, becomes worse, which is surprising. The paper argues that this is because of the lack of modeling the epistemic uncertainty. It then modifies the algorithms (PlaNet and Dreamer with appropriate inference) to add epistemic uncertainty following the Monte Carlo Dropout (MCD) approach but still obtains worse performance than the original performance of PlaNet and Dreamer.

The paper then postulates that the parameterization of the RSSM model is not suitable for smoothing (the appropriate inference technique), and proposed a new parameterization of the model using the linear-Gaussian model, VRKN. It shows that with VRKN and smoothing, in deterministic environments, both PlaNet and Dreamer match the performance achieved by the same algorithms with RSSM (original algorithms). In partial observable environments, VRKN (can not tell if MCD is used or not) is shown to be better than RSSM.

**Broader Impact Concerns:**

Not applied.

**Requested Changes:**

I would like the authors to think about my question and also discuss it in the paper (maybe some new experiments are needed to support your hypothesis) and also address the 7 weakness points mentioned above.

**Strengths And Weaknesses:**

I would like to first hear the authors' thoughts about the following question. Suppose we can find p and q such that inequality (1) becomes equality, when the agent wants to query the model for planning, it can not estimate the smoothing belief, which requires future information. Given that only the posterior is available anyway at decision time, why do we use the smoothing belief when learning the model? Could it be even worse than using the posterior for model learning (what RSSM does)? To think about the above question, maybe it is good to consider the following two questions first. What is the uncertainty in your state estimation? Where does it come from?

Strengths:
The problem that the paper considers is important novel and nontrival.

The authors' perspective in analyzing this problem is novel and promising to me.

The paper shows that the problem indeed happens in practice (Figure 7).

The solution method proposed by the paper seems to resolve the problem (Figure 7) well.

The limitations of the paper were acknowledged and discussed.

Relevant papers were cited.

Weakness:

1. Many math notations used to describe the proposed algorithm are used in a confusing way. I think the authors mix the notations of quantities to be estimated and their estimates. For example, p(z_{t+1} | z_t, a_t) is defined to be the transition probability of the latent states in Section 2 but is also used as a parameterized function modeling this transition probability in Section 3.2. Similarly, in Section 3.2, the authors define the latent observation model p(w_t | z_t), which is a definition of the generation process of observations from latent states. But the definition involves \sigma_t^w, which is the encoder's uncertainty estimator, a solution parameter instead of a problem parameter. Similarly, the "posterior state estimate p(zt|o≤t, a≤t)". Similarly, z1, z2, z3 in figure 3 are estimates instead of actual latent states.

2. Overall, I find the description of the proposed algorithm not complete and is thus difficult to follow. For example, it is not clear how the algorithm produces the estimate of the posterior of the latent state without future information. It is also not clear how learning works. The paper only says "We train all parts of the model jointly by maximizing Equation 3". But Equation 3 involves q and the algorithm description does not mention how it deals with q (e.g., how does the algorithm parameterize it). Also, Equation 3 involves expectations and the paper does not mention how to deal with them. It is also not clear how data were generated and how they were used in training.

3. Some important statements are not justified:

When presenting the requirements of the desired algorithm, the paper says
"In addition, such an architecture can naturally integrate multiple sensors that emit data at different frequencies
into a common latent state representation."
This sentence looks strange because up to this point, the paper has not discussed why integrating data from multiple sensors is important and why RSSM can not achieve this goal.

Similarly, the paper says in the requirements:
"Both should be computable by the same network architecture to avoid an over-parametrization and overfitting as a consequence."
This is again unjustified.

4. Some important terminologies/statements need clear definition:
Section 2 is called "Inference and Learning in State Space Models" but I don't see a definition/description of what "inference" and "learning" mean in your problem.

The paper needs to either define or cite other works for aleatoric and epistemic uncertainty.

"Traditionally, the independence assumptions of the generative model are also used for inference." What do the independence assumptions mean?

"As exact inference is intractable for most models of interest, we usually use approximate methods such as
variational inference. For a single sequence, ..." It seems that you are about to explain the variational inference method here, but you didn't say it. And this terminology is not used anymore in the paper.

5. A comparison between MCD-VRKN and MCD-RSSM in the experiments producing Figures 5 and 6 is important but is missing.

6. I don't find a discussion in the main text referring to Figure 4.

7. It is unclear how Smoothing-RSSM does inference when future information is not available.

---

> ### Author Response · Authors · 2022-08-25
> **Reply to Reviewer TTvU**
>
> We thank the reviewer for their constructive feedback and for acknowledging that our paper provides an interesting and promising analysis of a non-trivial problem. We are currently revising the paper, running additional experiments to address the concerns of all reviewers, and will provide an updated version of the paper in due time.
>
> We would now like to answer the posed questions and afterward address the raised concerns individually.
>
> > What is the uncertainty in your state estimation? Where does it come from?
>
> There are two sources of uncertainty in the state estimate:
> 1.) Lack of information about the current system state (aleatoric uncertainty). This uncertainty is usually caused by certain aspects of the environment being unobservable and/or inherently stochastic (e.g., transition noise). In particular, no amount of training data can resolve this uncertainty, it needs to be resolved during acting in the environment. In the standard DeepMind Control suite benchmarks, there is almost no noise in the dynamics and observation generation process, so the aleatoric uncertainty is very low. We constructed the modified benchmarks (occlusion and fusion experiments) to increase the aleatoric uncertainty and analyze how well the different models can handle this.
> 2.) Uncertainty about the model parameters (epistemic uncertainty), caused by limited training data. This form of uncertainty can be reduced by additional training data. This aspect is especially relevant for model-based RL applications where the agent collects the data on the fly. Especially at the beginning of training, there is little data, covering only a small part of the observation-action space.
> In our paper, we argue that distinguishing these cases during state-space modeling is important and provide a well-founded approach to addressing this issue.
>
> > Suppose we can find p and q such that inequality (1) becomes equality, when the agent wants to query the model for planning, it can not estimate the smoothing belief, which requires future information. Given that only the posterior is available anyway at decision time, why do we use the smoothing belief when learning the model? Could it be even worse than using the posterior for model learning (what RSSM does)?
>
> It is important to distinguish the model learning phase from using the model for planning or control. For training, first, note that inequality 1 can only be equality if we use a smoothing inference (the variational bound can be tight in this case). For this reason, it is well-established in traditional system identification that smoothed state beliefs are required to learn the correct model parameters, dating back to at least [1]. Yet, as we argue, in the case of model-based RL using the RSSM inference can be beneficial as it implicitly regularizes the model. This observation is highly counter-intuitive and requires analysis. We provide such an analysis in our work.
> As mentioned, for control only the posterior is available and thus we need an architecture that can effectively compute both, posterior and smoothed beliefs. We provide such an architecture with the VRKN by providing a strong inductive bias that allows filtering during control time without explicitly training for it: The smoothing based on the Rauch-Tung-Striebel equations does not introduce any additional parameters and reuses the state estimates computed during filtering. Thus, to produce reasonable smoothed estimates, the model also has to learn to filter.
>
> We will consider these aspects when revising our work.

---

> > ### Author Response · Authors · 2022-08-25
> > **Addressing the Weaknesses 1/2**
> >
> > > 1. Confusing notation
> > >
> >
> > We agree that our notation lacks some rigor in the mentioned parts and will improve this in the revised version.
> >
> > > 2. Overall, I find the description of the proposed algorithm not complete and is thus difficult to follow. For example, it is not clear how the algorithm produces the estimate of the posterior of the latent state without future information. It is also not clear how learning works. The paper only says "We train all parts of the model jointly by maximizing Equation 3". But Equation 3 involves q and the algorithm description does not mention how it deals with q (e.g., how does the algorithm parameterize it). Also, Equation 3 involves expectations and the paper does not mention how to deal with them. It is also not clear how data were generated and how they were used in training.
> > >
> >
> > The general concern about the description of our algorithm is shared with other reviewers, and we will heavily revise the corresponding section and clarify the algorithm. Nevertheless, we want to shortly address the posed questions:
> >
> > - Given the latent observations and learned transition model, the posterior latent states are computed using the (extended) Kalman-Filter equations.
> > - We parameterize q by the encoder and the transition model (which is shared with the generative model p)
> > - We approximate the expectations using single sample estimates, following [2, 3] and many other deep VI methods.
> >
> > > 3. Some important statements are not justified:
> > >
> > >
> > > When presenting the requirements of the desired algorithm, the paper says "In addition, such an architecture can naturally integrate multiple sensors that emit data at different frequencies into a common latent state representation." This sentence looks strange because up to this point, the paper has not discussed why integrating data from multiple sensors is important and why RSSM can not achieve this goal.
> > >
> > > Similarly, the paper says in the requirements: "Both should be computable by the same network architecture to avoid an over-parametrization and overfitting as a consequence." This is again unjustified.
> > >
> > - We will further emphasize the importance of fusion as an important task, where accurate estimation of aleatoric uncertainty matters, and the amenability of our approach to this problem in the revised paper. Additionally, we want to emphasize that we do not claim that the RSSM cannot achieve this but only that our architecture provides a more natural and principled solution that empirically works better.
> > - As mentioned earlier, we need both the posterior and smoothing estimates, the former for planning/control and the latter for training the model. While we agree that they do not necessarily need to be computed by the same network this is still highly desirable as it drastically reduces the model complexity and the loss (if a separate network for the posteriors would be used, the objective would need to be extended to explicitly train that network) - we will clarify in the revision
> >
> > > 4. Some important terminologies/statements need clear definition: Section 2 is called "Inference and Learning in State Space Models" but I don't see a definition/description of what "inference" and "learning" mean in your problem.
> > >
> >
> > We refer to the problem of estimating state beliefs given a model and observations as inference and to the problem of fitting the model parameters as training (as opposed to “inferring the model parameters”) - we will ensure the terms are defined and used consistently during revision.
> >
> > > The paper needs to either define or cite other works for aleatoric and epistemic uncertainty.
> > >
> >
> > We will properly define and reference these terms in the revision.
> >
> > > "Traditionally, the independence assumptions of the generative model are also used for inference." What do the independence assumptions mean?
> > >
> >
> > We refer to the independence (factorization) assumptions of the underlying graphical model, in this case, the hidden Markov assumptions underlying state space models.
> >
> > > "As exact inference is intractable for most models of interest, we usually use approximate methods such as variational inference. For a single sequence, ..." It seems that you are about to explain the variational inference method here, but you didn't say it. And this terminology is not used anymore in the paper.
> > >
> >
> > All state space models considered in our work use variational inference for inference and the stochastic gradient variational Bayes framework[4] for training. Yet, we agree that we mix these aspects in Section 2.1 and will more rigorously separate these during revision.

---

> > > ### Author Response · Authors · 2022-08-25
> > > **Adressing the Weaknesses 2/2**
> > >
> > > > 5. A comparison between MCD-VRKN and MCD-RSSM in the experiments producing Figures 5 and 6 is important but is missing.
> > > >
> > >
> > > We refer to the VRKN with Monte Carlo Dropout as just “VRKN” (as usage of MCD is the default for this model) and the baseline without MCD as “VRKN (no MCD)”. We compare the RSSM to its baselines (including the MCD-RSSM) in Figure 4 and the RSSM to the VRKN approaches in Figure 5. Note that the RSSM curves (green) are the same in both figures. For the modified benchmarks we did not yet evaluate the MCD-RSSM as their main focus is showing that appropriate estimation of aleatoric uncertainty improves performance if this form of uncertainty is important. The MCD-RSSM suffers from the same over-estimation problem as the normal RSSM. Additionally, those experiments are costly to run, and we did not see any performance differences between MCD-RSSM and RSSM in the previous experiments (Section 4.1, Figure 4). We will ensure this trend persists for the modified benchmarks on a subset of the experiments.
> > >
> > > > 6. I don't find a discussion in the main text referring to Figure 4.
> > > >
> > >
> > > This reference is indeed missing, and we will add it. We thank the reviewer for pointing it out. The figure corresponds to the first part of section 4.1 and provides evidence underpinning the discussion in section 2.3
> > >
> > > > 7. It is unclear how Smoothing-RSSM does inference when future information is not available.
> > > >
> > >
> > > We pass the current observation and initial cell state (all zeros) through the GRU cell. This is certainly not very elegant but follows from the design of the smoothing RSSM. We thus argue this naive modification is not sufficient and introduce the VRKN as an alternative. We agree that this is not clear from the current version of the manuscript and will elaborate.
> > >
> > > We once again thank the reviewer for their feedback and hope that we could adequately address all concerns.
> > > We invite them to further discussion if questions or other open points remain. As mentioned, we will provide a revised version of the paper in due time and ask for the reviewer's patience as appropriate incorporating all points and running the additional experiments will take us some time.
> > >
> > > [1] An Approach to Time Series Smoothing and Forecasting using the EM Algorithm, Shumway and Stoffer 1982,
> > >
> > > [2] Learning Latent Dynamics for Planning from Pixels, Hafner et al 2019
> > >
> > > [3] Dream to Control: Learning Behaviors by Latent Imagination, Hafner et al 2020
> > >
> > > [4] Auto-Encoding Variational Bayes, Kingma and Welling, 2014

---

### Review · Reviewer_5H8i · 2022-08-15

**Summary Of Contributions:**

The paper studies a popular type of state space models (SSM) used in recent model-based RL work: RSSMs. The authors point out an important limitation in the inference process (namely the lack of smoothing). The authors speculate this limitation has a beneficial effect on final agent performance through a regularization effect (namely overestimating aleatoric uncertainty), which means simply “fixing it” naively, in the current model architecture, degrades performance. The authors instead propose a different model architecture based on a Linear-Gaussian SSM where smoothing doesn’t require additional parameters (through a Kalman backward algorithm), that can still deal with high-res inputs, and where the beneficial regularization can be separately controlled with Monte Carlo Dropout.

The different approaches are tested with PlaNet and Dreamer agents on 3 Mujoco tasks. Some further analysis is done (e.g.  to check how the model deals with occlusion).

**Broader Impact Concerns:**

No concerns.

**Requested Changes:**

If my comment about sensor fusion above is justified, maybe soften the claims about sensor fusion.


**Strengths And Weaknesses:**

**Strengths**:

The paper is pleasant to read. Overall, it’s written clearly and it’s easy to follow as a result. I appreciated the background sections and the explanations around the effect of the approx inference process in Sec 2.2. The paper addresses potential shortcomings in a popular RL strand of work, the work combines many areas of expertise across approximate probabilistic inference, deep learning and model-based RL, and this is done in a way that is accessible to many.

The experiments are clear and illustrative, they generally support what is being described and the different hypotheses at play.

**Weaknesses**:

* Probably the main weakness of the paper is the limited gain that is demonstrated using the proposed model in the MuJoCo experiment. I think this is fine given the other contributions.

* The paper mostly focuses on the model architecture and inference, but there is little discussion of how this model is integrated in a control setting (the new proposed model is used as a drop-in replacement for the existing RSSM model).

* The authors have tried a simple change to integrate smoothing in the RSSM model (“smoothing RSSM”), which didn’t work well in practice. They then introduce the VRKN model, which is a big change from the existing model, and I wish there were more discussion of the design choices and consequences here (beyond just the smoothing aspect). For example, what is the additional cost to doing smoothing? Is it still a suitable model for RL / planning? e.g. Can simulated latent trajectories when planning be decoded to sensible observation trajectories? Is it more/less sensitive to action-conditioning?

* Sensor fusion: If I understood correctly, missing observations in the VRKN under the “natural” fusion mechanism means repeated observations. It seems unfair to compare it to RSSM with default values for unavailable observations. If the main positive effect here is through the observation repeat, then it’s not a terribly convincing demonstration of the fusion capability.

---
Other comment:

Another paper to consider citing in the space of SSMs for model-based RL: “Learning and Querying Generative Models for RL”

---

> ### Author Response · Authors · 2022-08-25
> **Reply to Reviewer 5H8i 1/2**
>
> We thank the reviewer for their constructive feedback, for the generally positive review, and for acknowledging that our paper presents a clear and empirically well-supported analysis of potential shortcomings in a popular RL method. We are currently revising the paper, running additional experiments to address the concerns of all reviewers, and will provide an updated version of the paper in due time. We would now like to address their concerns individually. We will also provide a revised version of the paper in due time after incorporating the feedback from all reviewers.
>
> > Probably the main weakness of the paper is the limited gain that is demonstrated using the proposed model in the MuJoCo experiment. I think this is fine given the other contributions.
> >
>
> We thank the reviewer for acknowledging that empirical improvements over the baselines are not everything that counts in research and nevertheless want to again emphasize that we see the core contribution of our work in deepening the understanding of SSM-based model-based RL methods and providing a path for future improvements. Given the popularity of RSSMs we assume this will be of interest to the community.
>
> > The paper mostly focuses on the model architecture and inference, but there is little discussion of how this model is integrated in a control setting (the new proposed model is used as a drop-in replacement for the existing RSSM model).
> >
>
>  We agree that the controller plays an important role and shortly point out the lack of discussion of those aspects in the limitations. Yet, we also believe that an additional study of control aspects is out of scope and that keeping the controller fixed for all approaches is the only way to a fair comparison of the underlying state space models in a model-based RL setting. Additionally, by using both the PlaNet-based and Dreamer-based agents in Section 4.1, we demonstrate that our findings are not exclusive to one specific controller.
>
> > The authors have tried a simple change to integrate smoothing in the RSSM model (“smoothing RSSM”), which didn’t work well in practice. They then introduce the VRKN model, which is a big change from the existing model, and I wish there were more discussion of the design choices and consequences here (beyond just the smoothing aspect). For example, what is the additional cost to doing smoothing? Is it still a suitable model for RL / planning? e.g. Can simulated latent trajectories when planning be decoded to sensible observation trajectories? Is it more/less sensitive to action-conditioning?
> >
>
> Further details and motivation of the VRKN were also requested by the other reviewers and will be provided in the revision. Still, we would like to address the asked questions shorty here:
>
> - While smoothing clearly introduces some additional cost the models’ runtime is largely determined by the convolutions in the encoder and decoder and the additional backward path adds minimal overhead. Empirically, during training, the VRKN is about 20 percent slower than the RSSM, due to its more involved architecture.
> - We evaluate the VRKN in a model-based RL setting and show its suitability, the PlaNet agents in particular use the VRKN for planning. Here we only decode the reward but decoding sensible observations is also possible. We are unsure if we understood this part of the question correctly, if this is not the case we would kindly ask the reviewer to elaborate.
> - We have not studied the sensitivity to action-conditioning and are unsure how such a study would look like, in particular as Model-based RL is not possible without action-conditioning. Note that we build our model on insights provided in [1] which studies the effects of action conditioning in great detail - we will point this out in more detail. Again, we would kindly ask the reviewer to elaborate if concerns remain.
>
> > Sensor fusion: If I understood correctly, missing observations in the VRKN under the “natural” fusion mechanism means repeated observations. It seems unfair to compare it to RSSM with default values for unavailable observations. If the main positive effect here is through the observation repeat, then it’s not a terribly convincing demonstration of the fusion capability.
> >
>
> We want to emphasize that all approaches get the same inputs: Observations if valid, default value if not, and 0/1 indicator which observations are valid and which are not. The VRKN can use those indicators better as the formulation using the Kalman Filter equations allows to explicitly ignore the default values while the RSSM has to learn to ignore - we consider this an inherent advantage of our approach whose usefulness we validate here. We will make this more explicit during the revision. In particular, the VRKN does not receive repeated observations for missing information but rather no information at all (other than the flag indicating that there is no information).

---

> > ### Author Response · Authors · 2022-08-25
> > **Reply to Reviewer 5H8i 2/2**
> >
> > Lastly, we will include the mentioned paper in our related work. We once again thank the reviewer for their feedback and hope that we could adequately address all concerns.  We invite them to further discussion if questions or other open points remain.  As mentioned, we will provide a revised version of the paper in due time and ask for the reviewer's patience as appropriate incorporating all points and running the additional experiments will take us some time.
> >
> > [1] Action-Conditional Recurrent Kalman Networks For Forward and Inverse Dynamics Learning, Shaj et al 2020

---

### Review · Reviewer_uvqA · 2022-08-18

**Summary Of Contributions:**

The paper investigates the inference scheme used by Recurrent State Space Models (RSSMs) mainly pointing out the flaws in the core assumption of RSSMs is that the belief is independent of future observations instead of the correct smoothing assumptions. **Key contributions** are claimed to be as follows:

1. The authors discuss the shortcomings in the inference scheme of RSSMs, in that they formalize this shortcoming and analyze the assumptions’ effect on model learning leading to overestimating the aleatoric uncertainty. This is shown empirically that overestimating the aleatoric uncertainty can lead to poor performance in settings where “correctly estimating it is important”.

2. They implement a naive approach of smoothing to show that even this does not help cope with the over-estimation of the aleatoric uncertainty.

3. The authors present VRKN as a solution; it uses a latent linear Gaussian parameterization, combining components for both aleatoric and epistemic

4. Introduce a bayesian treatment of the transition model’s parameters where the systems epistemic uncertainty is modelled using Monte Carlo Dropout. This results in:
- performance is comparable to RSSM-based agents in a deterministic environment
- performance is improved when the tasks required aleatoric uncertainty estimation.


**Broader Impact Concerns:**

To the best of my knowledge, there are not any immediate broader impact concerns.

**Requested Changes:**

With the paper currently as it stands, I am unfortunately not able to make a recommendation for acceptance. Here are specific adjustments which I believe will strengthen the work:

- Please add clearly and precisely what the key contributions of this work are. At present, the introduction has dense paragraphs and it is left to the reader to enlist these.

- Please cross-refer to each contribution which section is covering what. At present, it is hard to tell which section is covering formalism, etc.

- 5 seeds are not at all sufficient given that the performance gains are marginal especially when the performance remains at par with the baseline. It would be important to repeat these experiments for at least 10 seeds or more to solidify these claims.

- Please add error bars wherever missing.

- What do the error margins indicate where they are actually drawn? Can you please elaborate on this, especially given there is both aleatoric and epistemic uncertainty being dealt with and claimed about here?

- It is not exactly clear how experiments in section 4.3 relate to the general idea/message of the paper. The section tries to an argument to relate it to uncertainty estimation, but it seems a bit tenuous.

- The presentation of the VRKN method is not clear enough. Many design choices, such as introducing latent observations, are hardly motivated and are not well-situated enough with respect to other parts of the method (Figure 3b). I would encourage the authors to carefully attend to this section, by highlighting differences/similarities with previous work.

- When the authors say in conclusion that “we analyzed the independence assumptions underlying Recurrent State Space Models (RSSMs) and found they are theoretically sub-optimal. Yet, they implicitly regularize the model by causing an overestimated aleatoric uncertainty and are crucial to the RSSMs success in model-based RL.” It is not at all convincing to me that the current set of experiments demonstrates this sufficiently. Alternatively, is this formally proven somewhere? If yes, I encourage the authors can present this in a formal theoretical statement. It is unclear to me to what degree the authors position this as a contribution or rather more as motivation for the specific approach they construct.

- Further, the authors say “When trying to avoid this heuristic approach and use the correct assumptions while replacing the implicit regularization with a more explicit approach using epistemic uncertainty, we found a simple extension of the RSSM architecture is insufficient. Thus, we redesigned the model from the first principles, using an inductive bias, which is appropriate for smoothing. As a result, we propose the Variational Recurrent Kalman Network (VRKN) which builds on well-understood tools for modelling the aleatoric and epistemic uncertainties. It uses extended Kalman smoothing for exact inference in a latent space to capture aleatoric uncertainty and explicit models the epistemic uncertainty using Monte Carlo Dropout.”  I really struggle following this part of the story. If I understand this correctly, the authors mean here that “redesigned model from first principles” includes as its main features “Kalman smoothing” and “Monte Carlo dropout”. To me, these are like well-known techniques that perform well as heuristics for helping estimate uncertainty, and the fact that they would be positioned here as being part of “first principles” is quite confusing to me.

- While the sensor fusion section is interesting, and I believe in itself could be the main positioning. But I believe that is not the case. If this is the main focus of the paper, it should have been discussed during the formulation.

- It seems that a lot is happening in the paper but it definitely could be both written and presented with significantly more clarity than the current standing of the work.


**Strengths And Weaknesses:**

**Strengths**

1. Investigates widely used state space models in model-based RL approaches (Dreamer, PlaNet, and its successors) and formulates a core assumption and its weakness thereafter. This is done in two parts,
- the misspecification of the inference distribution is shown by spelling out the variational lower bound decomposition to the log-likelihood of the observations given the actions, to indicate that the assumption of RSSMs leads to no tightness in the lower bound which is a strong limitation.
- the effect of inference assumption on model learning has been demonstrated using a simple linear-gaussian state space model, with a state dimensionality of 4 and a ground truth generative model. Figure 2 shows the RSSM/SSM model learning results where evidence is provided that RSSM indeed has higher transition variance, higher model error, and underestimates the log state probability.

2. The paper identifies the issue in RSSM arises because the model aims to maximize the ELBO while being evaluated based on the agent’s reward, to address this naively the authors propose Monte Carlo Dropout to called smoothing RSSM. Claims to be corroborated empirically later on in the paper somewhere.

3. The proposed method VRKN (Variational Recurrent Kalman Networks) seems theoretically grounded, which allows for
- parameter-free smoothing using a locally linear state space model in a latent space, while still scaling to image-based control tasks.
- one network architecture that can compute both smoothed distributions for training and (filtered) posteriors for online control, to avoid an over-parametrization, and naturally integrate multiple sensors that emit data at different frequencies into a common latent state representation.

4. A diverse set of experiments that probe for specific questions/weaknesses of the proposed method

**Weaknesses**

**Overall** I find the paper is not very well written and it is not very clear when it comes to stating its contributions. The empirical results are not so convincing either, which I guess leads the authors to kind of position the paper like this approach is more justified than the current SOTA and achieves the same performance. This positioning kind of hinges on establishing the theoretical justification first and foremost, making it the main contribution. However, I fear this also may not be a particularly strong point of the paper. Section 2.1 is the most theoretical part, but the soundness and significance of the analysis are not that clear to me. The main conclusions are made in sections 2.2 and 2.3, but the authors seem to be more hand-wavy/marginal empirical justifications than actual formal theoretical results/solid empirical analysis with rigorous ablations. Here are more detailed comments:

1. Figure 1, it is not clear to me why there are **no concrete error bars** here considering there are 1000 sequences of length 50 each, ideally there should be multiple runs here. Maybe I am missing something. Can you please clarify? From what I understand so far, you are plotting the median. But even that requires error bars here. For instance in col 2, orange there is a lot of variances noticed.

2. The paper is **not very clearly written and organized**. The introduction is very dense and it is hard to read. The authors introduce Smoothing RSSM in Section 2.3 and then claim there that “it does not help to improve the Smoothing RSSM’s performance. From these results, we conclude that solely addressing the sub-optimal inference assumption is insufficient, but we also need to rethink the model’s parameterization.” Where is the evidence for this provided? There is no reference or result here until I go to further sections which are not even referenced here. This is hard to parse.

3. Considering the paper is very much ablation-focused, **only 5 seeds** is not at all sufficient given that the performance gains are marginal especially when the performance remains at par with the baseline. It would be important to test experiments for at least 10 seeds or more to solidify these claims.

4. In Figure 6, we would expect to see a greater difference in performance as the main point of the paper is that the overestimation of the aleatoric uncertainty is not precise enough in RSSMs. However, the difference in performance is pretty narrow. Would be nice to know a bit more about why the method proposed by VRKN does not seem to do so well.

5. Another weakness is the assumption of a Gaussian generative distribution p(ot|zt) with fixed variance. What happens when this is not true in more practical settings?

6. The presentation of the VRKN method is not clear enough. Many design choices, such as introducing latent observations, are hardly motivated and are not well-situated enough with respect to other parts of the method (Figure 3b).

---

> ### Author Response · Authors · 2022-08-25
> **Reply to Reviewer uvqA 1/4**
>
> We thank the reviewer for their detailed and constructive feedback, as well as for acknowledging that our work analyzes an important issue with RSSMs, proposes a theoretically grounded alternative, and analyzes its findings on a diverse set of experiments that probe specific aspects of the studied methods.  We are currently revising the paper, running additional experiments to address the concerns of all reviewers, and will provide an updated version of the paper in due time.  We would now like to address the raised concerns individually and will also provide a revised version of the paper in due time after incorporating the feedback from all reviewers.
>
> > I find the paper is not very well written and it is not very clear when it comes to stating its contributions.
> >
>
> > Please add clearly and precisely what the key contributions of this work are. At present, the introduction has dense paragraphs and it is left to the reader to enlist these.
> >
>
> > Please cross-refer to each contribution which section is covering what. At present, it is hard to tell which section is covering formalism, etc.
> >
>
> We will add an explicit contribution statement with references to the specific sections in the revised version and also improve the general structure and clarity. We believe our key contributions are as follows:
>
> 1. We analyze the assumptions underlying the RSSM and find that they, unlike the conventional smoothing inference in SSMs, ignore future observations when inferring state beliefs. Such an inference is suboptimal and has subtle effects on model learning, leading to an overestimated aleatoric uncertainty (Section 2.1 and Section 2.2).
> 2. We argue that, counterintuitively, this suboptimal inference is beneficial for the RSSMs performance as it addresses objective mismatch in a heuristic way. It takes the role of epistemic uncertainty, which
> is crucial for many other model-based RL approaches (Section 2.3). We evaluate several modifications to the RSSM to provide empirical evidence to support these hypotheses (Section 4.1).
> 3. We introduce the VRKN, an alternative to the RSSM, which provides a better inductive bias for smoothing inference (Section 3). We again show that a smoothing inference without additional measures yields suboptimal performance, yet, when combined with epistemic uncertainty, the
> VRKN ’s improved inductive bias allows it to close the performance gap on standard benchmarks
> (Section 4.1).
> 4. We show that VRKN -based agents improve performance in tasks where correct uncertainty estimation matters, i.e., tasks with partial observability or tasks requiring sensor fusion. (Section 4.2
> and Section 4.3)
>
> > The introduction is very dense and it is hard to read
> >
>
> We will rewrite the introduction to be less dense and more understandable in the revision.
>
> > The authors introduce Smoothing RSSM in Section 2.3 and then claim there that “it does not help to improve the Smoothing RSSM’s performance. From these results, we conclude that solely addressing the sub-optimal inference assumption is insufficient, but we also need to rethink the model’s parameterization.” Where is the evidence for this provided? There is no reference or result here until I go to further sections which are not even referenced here. This is hard to parse.
> >
>
> As mentioned in Section 2.3 the corresponding evidence is provided in Section 4.1, we will make this reference clear and also explicitly reference the corresponding figure (Figure 4).
>
> > The empirical results are not so convincing either, which I guess leads the authors to kind of position the paper like this approach is more justified than the current SOTA and achieves the same performance.
> >
>
> Given the widespread use of RSSM-based methods and lack of analysis and understanding of those models, we would like to emphasize that we believe providing such an analysis and deepening the understanding is a valuable contribution to the community, even without improving the SOTA on the standard benchmarks.
>
> > Figure 1, it is not clear to me why there are **no concrete error bars** here considering there are 1000 sequences of length 50 each, ideally there should be multiple runs here. Maybe I am missing something. Can you please clarify? From what I understand so far, you are plotting the median. But even that requires error bars here. For instance in col 2, orange there is a lot of variances noticed.
> >
>
> For the simple example considered in Figure 2, we only ran a single seed and presented the results due to the simplicity of the experiment and the clarity of the results. By now we rerun the experiment for 5 seeds and obtained standard errors, which are several orders of magnitude smaller than the differences between the approaches and thus hard to visualize. Still, we will update the figure and clarify what exactly we report. Please also note the log scale in the middle and right column, which visually emphasizes small oscillations of the model after convergence.

---

> > ### Author Response · Authors · 2022-08-25
> > **Reply to Reviewer uvqA 2/4**
> >
> > > Considering the paper is very much ablation-focused, **only 5 seeds** is not at all sufficient given that the performance gains are marginal especially when the performance remains at par with the baseline. It would be important to test experiments for at least 10 seeds or more to solidify these claims.
> > >
> >
> > > 5 seeds are not at all sufficient given that the performance gains are marginal especially when the performance remains at par with the baseline. It would be important to repeat these experiments for at least 10 seeds or more to solidify these claims.
> > >
> >
> > We would like to emphasize that while we only use 5 seeds per environment, in section 4.1 we evaluate 2 different control methods (Cross-Entropy planning using the approach form [1] and Latent Imagination + a parametric policy using the approach from [2]) and multiple environments (6 for PlaNet-based Agents, 8 for Dreamer-based agents). We draw our conclusions not based on a single environment/control method but on the aggregated results, i.e., 70 runs (5 seeds * 8 environments for Dreamer-based agents + 5 seeds * 6 environments for PlaNet-based agents). This form of evaluation is common in the field[1, 2], largely due to the significant computational cost of training (about 5k GPU hours for the results reported in Figures 4 and 5), which prevents us from running more seeds per environment. We will improve how we report the results (see also the “error bar” related concerns below) using the approach presented in [3].
> >
> > For the results in sections 4.2 and 4.3, we only used 4 environments, Dreamer-based agents, and fewer state space modeling approaches (i.e., only the VRKN and RSSM). Thus we will increase the number of seeds for the revised version.
> >
> > > Please add error bars wherever missing.
> > >
> >
> > > What do the error margins indicate where they are actually drawn? Can you please elaborate on this, especially given there is both aleatoric and epistemic uncertainty being dealt with and claimed about here?
> > >
> >
> > We thank the reviewer for pointing out that our current way of reporting the results does not indicate their significance. As mentioned, we will use the approach presented in [3] to report the average results with appropriate error bars.
> > Additionally, we would like to point the reviewer to Appendix D where we already provide additional quantitative results, including Box plots, tables with means and standard errors of the final performances for all conducted experiments, and the reward curves for additional environments, that are not included in the main part of the paper. As described in the introduction to the experiment section (Section 4), the already present error bars for evaluations of the individual environments indicate 25%-75% quantiles. We will ensure we properly explain our evaluation scheme in the revision.
> >
> > > In Figure 6, we would expect to see a greater difference in performance as the main point of the paper is that the overestimation of the aleatoric uncertainty is not precise enough in RSSMs. However, the difference in performance is pretty narrow. Would be nice to know a bit more about why the method proposed by VRKN does not seem to do so well.
> > >
> >
> > We would first like to mention that control under such heavy occlusions and transition noise is a rather hard task, and no approach we know of “does well” here (with the same number of environment interactions). Additionally, many other aspects, e.g. the exact controller, are important for performance but out of scope for this work. Yet, we believe the results still demonstrate that appropriate estimation of aleatoric uncertainties with the VRKN slightly helps with control and clearly leads to a better representation (see reconstructions in Figure 7 and supplement).
> > Further, we would like to highlight that the sensor fusion experiments probe the same aspects (importance of the quality of the aleatoric uncertainty estimation) and provide a much clearer performance improvement by the VRKN.

---

> > > ### Author Response · Authors · 2022-08-25
> > > **Reply to Reviewer uvqA 3/4**
> > >
> > > > When the authors say in conclusion that “we analyzed the independence assumptions underlying Recurrent State Space Models (RSSMs) and found they are theoretically sub-optimal. Yet, they implicitly regularize the model by causing an overestimated aleatoric uncertainty and are crucial to the RSSMs success in model-based RL.” It is not at all convincing to me that the current set of experiments demonstrates this sufficiently. Alternatively, is this formally proven somewhere? If yes, I encourage the authors can present this in a formal theoretical statement. It is unclear to me to what degree the authors position this as a contribution or rather more as motivation for the specific approach they construct.
> > > >
> > >
> > > The fact that the RSSM bound is not tight is formally discussed in Section 2.1. The effects of the suboptimality on the estimated aleatoric uncertainty are visualized with a thought experiment and toy example in Section 2.2. Yet, we agree that these parts can be clearer and better separated. We will address this during revision.
> > >
> > > The regularizing effect of this over-estimation is hard to analyze theoretically.
> > > We provide the general idea in Section 2.3 but postpone the experimental results to after the introduction of the VRKN and details of the experimental setup, to Section 4.1. We will make this reference clear. These experiments show that naively smoothing with the RSSM hurts its performance and MCD does not improve the performance (Figure 4). From these results, we draw the conclusion that the RSSM provides a poor inductive bias for smoothing and thus propose the VRKN. We again show that naively smoothing (, i.e., removing the implicit regularization through over-estimation) decreases performance, but this time the original performance can be recovered by modeling epistemic uncertainty.
> > >
> > > We believe this argumentation sufficiently underpins our claims and hypotheses about the RSSM's workings (after we address the error bar and seed-related issues discussed above).
> > >
> > > > Further, the authors say “When trying to avoid this heuristic approach and use the correct assumptions while replacing the implicit regularization with a more explicit approach using epistemic uncertainty, we found a simple extension of the RSSM architecture is insufficient. Thus, we redesigned the model from the first principles, using an inductive bias, which is appropriate for smoothing. As a result, we propose the Variational Recurrent Kalman Network (VRKN) which builds on well-understood tools for modeling the aleatoric and epistemic uncertainties. It uses extended Kalman smoothing for exact inference in a latent space to capture aleatoric uncertainty and explicit models the epistemic uncertainty using Monte Carlo Dropout.” I really struggle following this part of the story. If I understand this correctly, the authors mean here that “redesigned model from first principles” includes as its main features “Kalman smoothing” and “Monte Carlo dropout”. To me, these are like well-known techniques that perform well as heuristics for helping estimate uncertainty, and the fact that they would be positioned here as being part of “first principles” is quite confusing to me.
> > > >
> > >
> > > We agree that the wording “first principles” is misleading here and will rephrase it. Additionally, we would like to emphasize that we do not believe Kalman Filtering and Monte Carlo Dropout are mere heuristics but that both techniques are theoretically well-motivated and established in the literature. In contrast, we consider replacing epistemic uncertainty with over-estimated aleatoric uncertainty a heuristic, as there is no theoretical understanding of how these uncertainties relate to each other and how we can control them using the given loss function.
> > >
> > > > The presentation of the VRKN method is not clear enough. Many design choices, such as introducing latent observations, are hardly motivated and are not well-situated enough with respect to other parts of the method (Figure 3b). I would encourage the authors to carefully attend to this section, by highlighting differences/similarities with previous work.
> > > >
> > >
> > > This concern is a shared concern among reviewers. Thus, we will heavily revise the corresponding section of the paper, addressing the raised issues while improving the motivation and explanation of the VRKN.

---

> > > > ### Author Response · Authors · 2022-08-25
> > > > **Reply to Reviewer uvqA 4/4**
> > > >
> > > > > It is not exactly clear how experiments in section 4.3 relate to the general idea/message of the paper. The section tries to an argument to relate it to uncertainty estimation, but it seems a bit tenuous.
> > > > >
> > > >
> > > > > While the sensor fusion section is interesting, and I believe in itself could be the main positioning. But I believe that is not the case. If this is the main focus of the paper, it should have been discussed during the formulation.
> > > > >
> > > >
> > > > While we agree that the fusion aspects are interesting for future research, we do not see them as the main positioning of this work. Yet, because these aspects are interesting and relevant we wanted to point out the amenability of our formulation to sensor fusion as an additional benefit.
> > > >
> > > > Additionally, the corresponding experiments (Section 4.3) are of interest as sensor fusion and also the image-only baselines (i.e., the images arrive at varying intervals) are good examples of tasks that require accurate estimates of the aleatoric uncertainty in each of the observations and the belief state itself. Only with such knowledge, it is possible to correctly integrate the information from all of these sources and when to rely on either of the sensors or the belief state.
> > > >
> > > > > Another weakness is the assumption of a Gaussian generative distribution p(ot|zt) with fixed variance. What happens when this is not true in more practical settings?
> > > > >
> > > >
> > > > No part of the derivations, discussion, or other modeling aspects relies on this assumption. The variance can easily also be parameterized by the decoder, which lifts the assumption if it is empirically beneficial. Yet, when working with image-based outputs, as we do, working with fixed variances is common, and we follow the design choices of [1, 2] as we directly compare with those methods.
> > > >
> > > > We again thank the reviewer for their feedback and hope we adequately addressed their concerns. We invite them to further discussion if questions or other open points remain. As mentioned, we will provide a revised version of the paper in due time and ask for the reviewer's patience as appropriate incorporating all points and running the additional experiments will take us some time.
> > > >
> > > > [1] Learning Latent Dynamics for Planning from Pixels, Hafner et al 2019
> > > >
> > > > [2] Dream to Control: Learning Behaviors by Latent Imagination, Hafner et al 2020
> > > >
> > > > [3] Deep Reinforcement Learning at the Edge of the Statistical Precipice, Agarwal et al 2021

---

> > > ### Comment · Action_Editors · 2022-08-29
> > > **Lack of compute**
> > >
> > > Too few seeds is indeed common in the literature, but in the end not a valid justification. There are many moving parts here, but generally speaking one needs to be very careful when estimating variability of the underlying performance distribution from few seeds.
> > >
> > > The paper states:
> > > >"We report the mean reward over all environments to show general trends and reward curves for the individual environments, showing median performance, where the shaded areas indicate 25% to 75%-percentiles. We use 5 seeds for each agent-environment pair, train for 1 million environment steps, and use 10 rollouts for evaluation."
> > >
> > > This suggests you might have 50 total samples for generating the shaded region. This could be why the shaded regions in the plots are so tight. But, they certainly aren't i.i.d. samples. You would need to use different statistical tools like a sign-test.
> > >
> > > You are reporting variability instead of uncertainty (for the shaded regions), I would have expected wider shaded regions regardless of number of samples. It could be due to the non-independence of the samples (this seems likely), which must be treated carefully. It could also be that the dist is pretty concentrated around the median with long tails (which are chopped off due to 25th percentiles instead of something more sensitive like 5th percentile).
> > >
> > > **Finally on compute.** Just because we don't have access to google-scale compute does not excuse experiments that don't support our claims and not well chosen statistical analysis. This is a bit like saying "my proof is not complete but I didn't have access to a theory guy, so just trust me that its true." In that case and with experiments we can only ask questions for which we have resources (compute and time) to evaluate properly.
> > >
> > > **Action items:**
> > > - in small domains like acrobat you can certainly run more seeds. Please do so.
> > > - in larger domains where you cannot run enough seeds and/or get significance: check the text of the paper. In such case we cannot claim SOTA or significant improvement.
> > > - please explain precisely how you are computing the shaded regions (give the algorithm ideally). We cannot average over environments easily so we have to make sure the procedure is correct and the statistical assumptions made are valid.

---

> > > ### Comment · Reviewer_uvqA · 2022-08-30
> > > **Response to 2/4**
> > >
> > > Thanks for the detailed response. I believe that the action editor (AE)'s response further elaborates my concerns about the lack of seeds and therefore the concerns about the author's statements on empirical findings. I encourage the authors to consider concrete action items enlisted there. Additionally when the authors say.
> > >
> > > >We would like to emphasize that while we only use 5 seeds per environment, in section 4.1 we evaluate 2 different control methods (Cross-Entropy planning using the approach form [1] and Latent Imagination + a parametric policy using the approach from [2]) and multiple environments (6 for PlaNet-based Agents, 8 for Dreamer-based agents). We draw our conclusions not based on a single environment/control method but on the aggregated results, i.e., 70 runs (5 seeds * 8 environments for Dreamer-based agents + 5 seeds * 6 environments for PlaNet-based agents).
> > >
> > > Something important here I want to emphasize is the distinction between the runs and environments. I understand the sentiment here is to show *many diverse* environments, but this calculation here (i.e. of "70 runs") is not reflecting anything about the significance of the results.  By any means, we cannot substitute runs per environment with the number of environments and do the math as suggested here ("70 runs").

---

### Author Response · Authors · 2022-08-30
**Addressing Significance of Results**

We thank the reviewer and AE for pointing out the issues with reporting our results and that their significance is unclear from the current plots. We rethought the reporting and decided to use the methodology recently proposed by Agarwal et al [1] which focuses on how to establish statistically significant results in Deep RL. Following their suggestions we will report the following metric:

- The **interquartile mean** of the average return (i.e., mean over the 10 rollouts) across all tasks and seeds.
- We will indicate uncertainty in results by **95% stratified bootstrap confidence intervals**, across all tasks and seeds.
- Note that the rewards in the Deep Mind Control Suite we use for evaluation are normalized between [0,1] for all environments and the rollouts have equal lengths (1000 steps), thus the average returns are normalized between [0. 1000]. This is also the case for the modified versions of the benchmarks, as we leave the reward untouched.  In this case, aggregating across tasks is recommended according to [1] to get better estimates.
- We will increase the **number of seeds to (at least) 10** for all tasks and make it more explicit that we base our claims on the aggregated results and not on the individual tasks in the writing. Further, we will move the results for the individual tasks to the supplement.
- We will use the library provided by [1] ( https://github.com/google-research/rliable ) and reference to both the paper and code to explain how the error bars are computed in the revised paper.

We kindly ask the reviewers and the AE for additional patience until we finished running the experiments (approx 2 weeks) and if they see any additional issues or have concerns with the suggested strategy.

[1]  Deep Reinforcement Learning at the Edge of the Statistical Precipice, Agarwal et al, NeurIPS 2021

---

### Author Response · Authors · 2022-09-14
**Revision**

We again thank the reviewers and action editor for their valuable feedback and have prepared a revised version. The revision incorporates the feedback and the points mentioned in our original responses. We marked the revisions in blue in the manuscript.
Furthermore, we provide a changelog containing the major points below.

Please let us know if there are any remaining concerns or open questions.

## Change Log:

**Section 1** **Introduction:**

- Added explicit contribution statement with references to individual parts of the paper (uvqA)
- Made motivation for sensor fusion experiments and the need for aleatoric uncertainty estimation more explicit (all)
- defined and cited aleatoric and epistemic uncertainty in the introduction (TTvU)
- Made introduction less dense (uvqA)

**Section 2 Inference and Learning in State-Space Models:**

- Restructured Sections 2.1 and 2.2 to clearly distinguish the discussions regarding tightness of the bounds and model learning. We also elaborated on the tightness of bound aspects, making it more formal and precise. ( TTvU, uvqA)
- Figure 2: Averaged over 10 seeds and added 95 % bootstrapped CIs, yet they are too small to be visible (uvqA)
- Clearer pointers to experiments from Section 2.3

**Section 3 VRKN:**

We rewrote large parts of this section

- Clarified and motivated desiderata of VRKN (TTVU) and design choices (uvqA)
- Better related model to the aspects discussed in Section 2
- Elaborated on:
    - Role of intermediate observation representation “w” and encoder’s uncertainty.
    - How to run inference and cost of smoothing inference
    - Training procedure and why it ensures reasonable posterior estimates.
    - How to use the model for online control.

**Section 4 Experiments:**

- Averaged all results over 10 seeds. Reported the aggregated results (over different tasks) using the procedure and metrics recommended by Argawal et al 2021. Made clear that we draw our conclusions based on those results. We can now show (95%CI) significance wherever we claim it (note that for some of the experiments in Section 4.1 the conclusion is that specific approaches work the same) (uvqA, Action Editor)
- Split clearer into “Experiments underpinning our discussion in Section 2” (4.1) and “Experiments that show when aleatoric uncertainty matters” (4.2)
- Added second occlusion experiment (Wall Occlusions)
- Put Missing Observation Experiments in a separate section and elaborated on them. They were previously only used as a baseline in the fusion experiment and addressed only in very few sentences. Yet, we believe this experiment again highlights the importance of capturing aleatoric uncertainty and the VRKN’s benefits. Thus, we decided to highlight it more. We also added a baseline addressing the concerns of reviewer 5H8i about the handling of missing observations ( VRKN (Cat).
- clarified how missing observations are handled by VRKN and RSSM, (for both “missing observations” and “fusion” sections), see also appendix C.3 (5H8i)

**Section 5 Related Work**

- added Buesing et al, 2018 (5H8i)

**Section 6 Conclusion**

- clarified claims in conclusion (uvqA)

**Supplement**

- Added Details about RSSM baselines (Appendix C.1 / C.3, TTvU, 5H8i)
- Per task results are now also reported using interquartile mean and 95% bootstrapped CIs (again c.f. Argawal et al 2021)

---

### Decision · Action_Editors · 2022-09-25

**Recommendation:** Accept as is

**Comment:**

Three knowledgeable reviews all agree on acceptance. The initial submission had issues with the empirical setup and the authors went to great lengths to address these issue along with addressing the clarity concerns raised by the reviewers.